# Continuous monitoring of surface water vapour isotopic compositions at Neumayer Station III, East Antarctica

Saeid Bagheri Dastgerdi[1], Melanie Behrens[1], Jean-Louis Bonne[1], Maria Hörhold[1], Gerrit Lohmann[1], Elisabeth Schlosser[2,3], and Martin Werner[1]

[1]Alfred Wegener Institute Helmholtz-Center for Polar and Marine Research Bremerhaven, Bremerhaven, Germany
[2]Department of Atmospheric and Cryospheric Sciences, University of Innsbruck, Innsbruck, Austria
[3]Austrian Polar Research Institute, Vienna, Austria

**Correspondence:** Saeid Bagheri Dastgerdi (saeid.bagheri@awi.de)

**Abstract.** In this study, the first fully-continuous monitoring of water vapour isotopic composition at Neumayer Station III, Antarctica, during the two-year period from February 2017 to January 2019 is presented. Seasonal and synoptic-scale variations of both stable water isotopes $H_2^{18}O$ and $HDO$ are reported, and their link to variations of key meteorological variables are analysed. In addition, the diurnal cycle of isotope variations during the summer months (December/January 2017/18 and
2018/19) has been examined. Changes in local temperature and specific humidity are the main drivers for the variability of $\delta^{18}O$ and $\delta D$ in vapour at Neumayer Station III, both on seasonal and shorter time scales. In contrast to the measured $\delta^{18}O$ and $\delta D$ variations, no seasonal cycle in the Deuterium excess signal ($d$) in vapour is detected. However, a rather high uncertainty of measured $d$ values especially in austral winter limits the confidence of this finding. Overall, the $d$ signal shows a stronger inverse correlation with specific humidity than with temperature, and this inverse correlation between $d$ and specific humidity is
stronger for the cloudy-sky conditions than for clear-sky conditions during summertime. Back trajectory simulations performed with the FLEXPART model show that seasonal and synoptic variations of $\delta^{18}O$ and $\delta D$ in vapour coincide with changes in the main sources of water vapour transported to Neumayer Station. In general, moisture transport pathways from east lead to higher temperatures and more enriched $\delta^{18}O$ values in vapour, while weather situations with southerly winds lead to lower temperatures and more depleted $\delta^{18}O$ values. However, for several occasions, $\delta^{18}O$ variations linked to wind direction changes
were observed, which were not accompanied by a corresponding temperature change. Comparing isotopic compositions of water vapour at Neumayer Station III and snow samples taken in the vicinity of the station reveals almost identical slopes, both for the $\delta^{18}O$–$\delta D$ relation and for the temperature–$\delta^{18}O$ relation.

## 1 Introduction

During the last decades, Antarctic ice cores have been proven to be one of the most important climate archives. They have
enabled a detailed climate reconstruction on glacial-interglacial time scales over the last 800,000 years (EPICA Members, 2004; Jouzel et al., 2007) and are furthermore the only climate archive which allows direct measurements of past greenhouse gas changes (Petit et al., 1999; Siegenthaler et al., 2005; Loulergue et al., 2008; Lüthi et al., 2008). Recent Antarctic ice core

records also allow studying the phasing of climate changes and the linkage between northern and southern polar regions in an unprecedented way (WAIS Divide Project Members, 2015).

Measurements of stable water isotopes (here given in the usual $\delta^{18}O$ and $\delta D$ notation) are crucial for the analyses of ice core records. Since the pioneering work of Dansgaard (1964), Lorius and Merlivat (1975), and others, stable water isotopes in Antarctic ice cores have been used as a proxy to reconstruct past temperature changes. Recent intensive sampling campaigns on various expeditions and locations in Antarctica have increased our knowledge about the present-day variability of $\delta^{18}O$ and $\delta D$ at different spatial scales (Masson-Delmotte et al., 2008; Münch et al., 2016; Casado et al., 2017). Furthermore, measurements

of stable water isotopes of Antarctic deep ice cores revealed insights into temperature changes on glacial-interglacial time scales (e.g. Jouzel and Merlivat, 1984).

The fundamental physical processes, which link the isotopic fractionation processes during precipitation formation in clouds, the condensation temperature, and the surface temperature at the precipitation location, are well understood (Jouzel and Merlivat, 1984). However, in the interior of Antarctica, up to 60% of annual precipitation can be diamond dust or so-called clear-sky

precipitation, and hoar frost, which are not related to clouds (Walden et al., 2003; Stenni et al., 2016). Also, the quantification of the temperature–isotope relationship remains difficult. While early studies have assumed an equality between the observed modern spatial temperature–isotope relation and the temporal relation required to reconstruct past temperatures (Petit et al., 1999), it has become clear that the spatial relationship may differ from the temporal relationship (e.g. Sime et al., 2009). A proper isotope paleothermometer calibration is hampered by different processes, like changes in the temperature inversion

strength (Van Lipzig et al., 2002), changes in the seasonality or intermittency of snowfall events (e.g. Masson-Delmotte et al., 2006), changes in the glacial ice sheet height (Werner et al., 2018) and sea ice extent (Noone and Simmonds, 2004), or changes in the origin and transports pathways of moisture to Antarctica (e.g. Masson-Delmotte et al., 2011; Sime et al., 2013).

Another process that might alter the $\delta^{18}O$ and $\delta D$ value in Antarctic ice cores is the post depositional exchange of water isotopes between vapour and the surface snow layer. Since a few years, new commercial laser instruments have enabled

continuous in situ measurements of atmospheric water vapour isotopes with high sampling frequency (Kerstel et al., 1999; Lee et al., 2005). Recent studies on the Greenland ice sheet have reported an isotopic exchange between the near-surface vapour and the surface snow between precipitation events (Madsen et al., 2019; Steen-Larsen et al., 2014a). For the location of Dome C, East Antarctica, Casado et al. (2018) also concluded from a combination of precipitation and snow pit samples that the water isotopic composition of ice cores is not solely governed by the $\delta$-signal in precipitation. So far, only a few observational studies

directly monitoring vapour isotopic composition have been performed in Antarctica. They have all been limited to relatively short measurement periods during the austral summer season. Ritter et al. (2016) carried out continuous measurements of water vapour isotopes at Kohnen Station, East Antarctica, during the austral summer season 2013–2014. Their isotopic data show a similar pattern for different days with a strong diurnal cycle. Casado et al. (2016) performed measurements of water vapour isotopes in the austral summer season 2014–2015 at Concordia Station on the East Antarctic Plateau. They found two different

patterns of diurnal isotope cycles. One pattern showed almost stable water vapour isotopes, while the second pattern illustrated a very high correlation between water vapour isotopes, specific humidity, and temperature. Both studies by Ritter et al. (2016) and Casado et al. (2016) focused their analyses on the diurnal cycle and showed that the isotopic composition in Antarctic snow

changes during periods without precipitation. Bréant et al. (2019) measured the isotopic composition of water vapour during a 40-day period in the austral summer season 2016–2017 at the Dumont d'Urville station in Adélie Land. They also report clear diurnal cycles of temperature, specific humidity and isotopic composition. Their measurements and analyses showed that low $\delta^{18}O$ and high $d$ values in vapour at Dumont d'Urville could be linked to katabatic winds, which transport strongly depleted vapour from the inner East Antarctic ice sheet to the station. In contrast, Kurita et al. (2016) used shipboard observations of the isotopic composition of water vapour between Australia and the East Antarctic coast to analyse the influence of marine air intrusion on temporal isotopic variations in vapour at Syowa Station, East Antartica, during the austral summer seasons 2013–14 and 2014–15. They showed that northerly winds, associated with cyclones, might push marine air with isotopically enriched moisture into the East Antarctic inland.

Here, we report measurements of the isotopic composition of vapour from the Neumayer Station III, Antarctica, performed over a full two-year period between February 2017 and January 2019. To our knowledge, this is the first study continuously monitoring Antarctic vapour during all seasons of the year. The primary purpose of this study is a characterization of water vapour and related isotope changes at Neumayer Station III on annual, seasonal, and sub-seasonal time scales. This dataset can form the basis for further process studies on the potential vapour–snow exchange of stable water isotopes in Antarctica. In addition, it can be used as a new dataset to evaluate the capability of global and regional climate models with explicit water isotope diagnostics (e.g. Risi et al., 2010; Werner et al., 2011; Pfahl et al., 2012) to correctly simulate mean isotopic composition and temporal variations of $\delta^{18}O$ and $\delta D$ in Antarctic water vapour.

This paper is organized in five sections. The following Section 2 explains our instrumental setup to perform continuous measurements of water vapour and its isotopic composition at Neumayer Station III. In this section, other meteorological datasets and methods used in this study are described, too. In Section 3, we report on the performed measurements and analyses, which focus on the relation between changes of water isotopes in vapour and several key meteorological variables on different time scales. Section 4 discusses our results in a broader context and compares them to the reported isotope variations in vapour at other Antarctic locations. The results of this study will be summarized in the final Section 5.

## 2 Methods and data

### 2.1 Study site

Neumayer Station III (simply called Neumayer Station, hereafter) is a German Antarctic research station, located at $70°40'$S, $8°16'$W (Fig. 1) run by the Alfred Wegener Institute (AWI), Helmholtz-Centre for Polar and Marine Research. AWI has provided continuous synoptic observations since 1981, first at the former Georg-von-Neumayer Station ($70°37'$S, $8°22'$W) until March 1992, thereafter at Neumayer Station II ($70°39'$S, $8°15'$W) and since February 2009 at Neumayer Station III ($70°40'$S, $8°16'$W) (König-Langlo and Loose, 2007). The station is situated on the 200-meter thick Ekström ice shelf approx. 42 m above sea level. This ice shelf has a homogeneous, flat surface slightly sloping upwards to the south (Klöwer et al., 2013). While the shortest distance from the station to the ice shelf edge amounts to 6.2 km (towards ENE), the base is moving with the ice shelf towards the open sea in the north (16 km distance) at about 200 meters per year (König-Langlo and Loose, 2007).

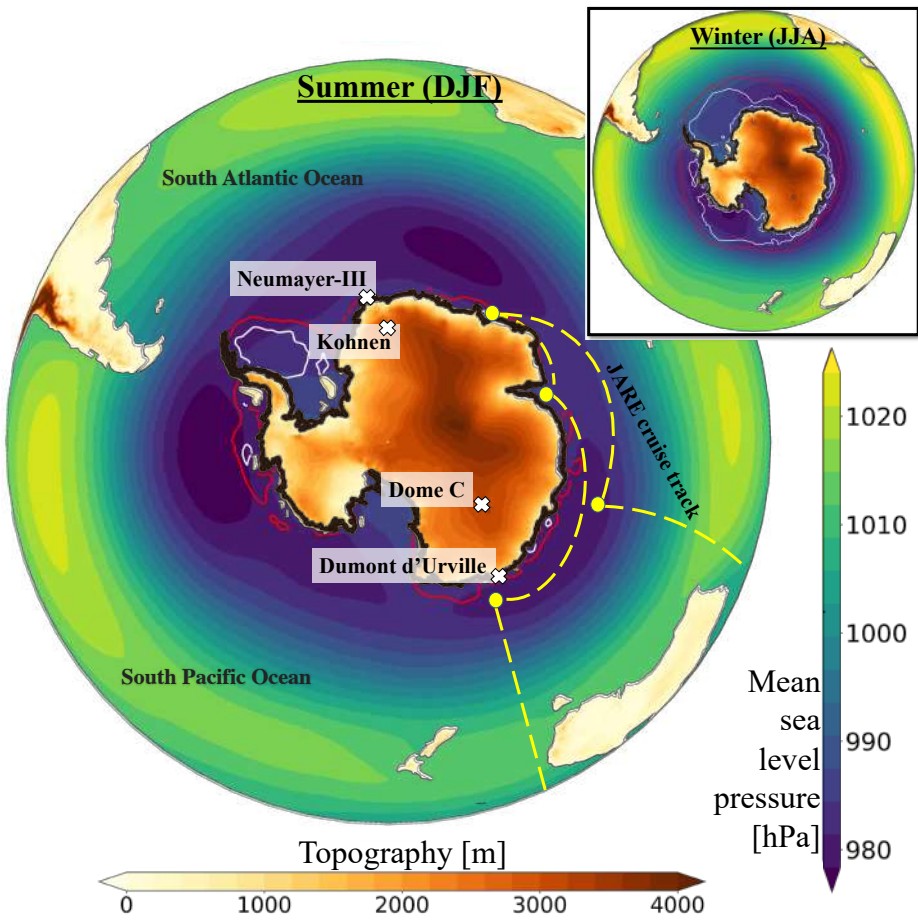

**Figure 1.** Map of Antarctica with topography [meter], mean sea level pressure [hPa], Antarctic grounding line (black line), and sea ice fraction [red line: fraction $> 0.45$, white line: fraction $> 0.90$] in austral summer (big map) and austral winter (small map), considering years of 2017 and 2018. The topography, mean sea level pressure and sea ice fraction are based on meteorological data from the European Centre for Medium-range Weather Forecasts (ECMWF) ERA5 reanalysis (Hersbach et al., 2020) and the Antarctic grounding line is based on Depoorter et al. (2013). The location of our study (Neumayer III) and other studies which provided continuous water vapour isotopic measurements in Antarctica (Ritter et al., 2016; Casado et al., 2016; Bréant et al., 2019) are shown in white colour and JARE cruise track related to Kurita et al. (2016) is shown in yellow colour.

Close to Neumayer Station, the sea ice extent has a minimum in February and a maximum in September and the coastlines are usually ice-free for parts of the summer (König-Langlo and Loose, 2007).

Like most coastal Antarctic stations, weather and climate at Neumayer Station are characterized by relatively high wind speeds, with an annual mean value of 8.7 $m\ s^{-1}$ (1-sigma standard deviation of calculated mean value from daily values: 95 $\pm 5.7 m\ s^{-1}$). Large wind direction and wind speed variations at Neumayer Station reflect complex dynamical processes re-

sulting from travelling cyclones around the station and katabatic winds (Kottmeier and Fay, 1998). There are two main wind directions. The prevailing wind direction is from east, caused by the passage of cyclones north of the Antarctic coast in the circumpolar trough. These low-pressure systems move toward west around Antarctica. Easterly storms are common, with wind speeds up to approx. 40 $m\ s^{-1}$. The second, less frequent, typical wind direction is south to south-west, caused by a mixture of weak katabatic and synoptic influence, with typical wind speeds below 10 $m\ s^{-1}$ (König-Langlo and Loose, 2007; Rimbu et al., 2014).

At Neumayer Station snowfall events usually coincide with strong winds which lift up surface snow from the ground into the air at the same time. Thus, the snow in the air close to surface is a mixture of precipitation and old surface snow, called blowing snow (Schlosser, 1999). Depending on the snow surface conditions, drifting or blowing snow can start at relatively low wind speeds between 6 $m\ s^{-1}$ and 12 $m\ s^{-1}$. At Neumayer Station, drifting or blowing snow is reported in 40 % of all visual observations (König-Langlo and Loose, 2007). Direct quantification of precipitation is almost impossible due to the influence of drifting and blowing snow. The mean annual accumulation has been determined glaciologically and amounts to approx. 340mm w.e. (water equivalent). The mean annual temperature at Neumayer Station (since 1981) is $-16.1 \pm 1.1°C$ (the uncertainty indicates the 1-sigma standard deviation of annual mean temperatures from the long-term mean, calculated for the period 1981-2018). Since the beginning of the measurements in 1981 no significant temporal trend in air temperature has been observed (Medley et al., 2018).

## 2.2 Meteorological observations

In this study, temperature, relative and specific humidity, wind speed, and wind direction data, measured 50 meters away from the station at 2-meter height above the surface, are used. We use meteorological observations at a resolution of 1 hour for the period February 2017-January 2019 (Schmithüsen et al., 2019). The climatology of temperature, specific humidity, and relative humidity is defined as the multi-year daily average value and the standard deviation over the 38-year period from 1981 to 2018 (König-Langlo, 2017). To identify specific days with extreme high or low temperature we compare the daily temperature at the station with its climatology. A day is considered as a warm (cold) event, when its temperature is at least one standard deviation above (below) the climatology (Klöwer et al., 2013).

## 2.3 Water vapour isotopic observations

In January 2017, a PICARRO L2140-i cavity ring-down spectroscopy analyser (simply named Picarro analyser, hereafter), has been installed in a laboratory room at Neumayer Station. The Picarro analyser is running continuously and measures the injected water vapour content and its isotopic composition approx. every two seconds. For our analyses, the data are reported as hourly mean values, if not stated otherwise.

For most of the year, wind at Neumayer Station blows from easterly, southerly, or south-westerly directions. Therefore, the air inlet for the vapour measurements is located on eastern side of the roof towards the mean wind direction and at its southern end to avoid exhaust gases. Apart from the time when the instrument is calibrated (see below), ambient air is constantly transported by an electrical pump from the rooftop to the Picarro analyser. During the calibration, to avoid keeping old air

inside the tube, the pump continues sucking the air from the rooftop without sending it to the Picarro analyser. The whole inlet tube (approx. 10m long) is constantly heated to approx. 65°C to avoid any condensation of vapour within the inlet tube.

Part of the instrumental setup is a custom-made calibration system using three different isotopic water standards (liquid), with $\delta^{18}O$ values of $-6.07 \pm 0.1$ ‰ (around $-17$ ‰ in water vapour), $-25.33 \pm 0.1$ ‰ (around $-36$ ‰ in water vapour), and $-43.80 \pm 0.1$ ‰ (around $-54$ ‰ in water vapour). $\delta D$ values of the standards (water liquid) are $-43.73 \pm 1.5$ ‰, $-195.21 \pm 1.5$ ‰, and $-344.57 \pm 1.5$ ‰. If evaporation during calibration is taken into account (see below), the chosen isotope values of the standards cover the whole range of expected isotope values in vapour at Neumayer Station (from -17 ‰ to -54 ‰ for $\delta^{18}O$ and from -120 ‰ to -404 ‰ for $\delta D$) during the course of a year. Isotope standards are provided in liquid form, and a bubbler system similar to the one described in Steen-Larsen et al. (2013) is used for vapourizing and measuring the standards. For safety reasons, two independent cooler boxes with two isotopic standards, each, are installed, and the isotope standard with a $\delta^{18}O$ value of -25.33 ‰ is measured in both cooler boxes. The boxes are held at a constant temperature of 17°C and air and water temperatures inside each bubbler system are constantly logged to determine the isotopic value of the vapour stemming from the liquid standards during the calibration measurements.

For calibrating our isotope measurements, the calibration protocol developed by Steen-Larsen et al. (2013) and Bonne et al. (2014) has been applied and modified. The calibration procedure includes (i) the correction of isotope measurements at low water concentration by determining the required humidity response functions of the Picarro analyser, (ii) corrections of a potential long-term drift of the instrument, (iii) the correction for an offset between measured and real isotope values, and (iv) filtering the data of special events, -e.g. weather conditions with a potential contamination of our vapour measurements by the station's exhaust gases, or days with temperature stabilization problems in the cooler boxes.

The range of water concentration defined for the Picarro analyser is 1000 to 50000 ppm (parts per million expressed by volume/volume), which equals specific humidity values in the range of 0.62 to 31.10 $g\ kg^{-1}$). At Neumayer Station, water concentration easily reaches values below 1000 ppm in the austral winter. For water concentrations lower than 2000 ppm (specific humidity of 1.24 $g\ kg^{-1}$), the analyser shows systematic errors with biases of more than 1 ‰ for $\delta^{18}O$ (Casado et al., 2016). To correct these systematic errors, we need to assess humidity response functions for our Picarro analyser. The Humidity response functions for all four isotope standards are determined once a year. Isotope values for water concentration ranging between 100 ppm and 10000 ppm (specific humidity between 0.06 and 0.62 $g\ kg^{-1}$) are measured several times and a best-fitting curve ($2^{nd}$ degree polynomial) is calculated. We do not find any change in the fitting curve between the different years of calibration. For determining and correcting an instrumental drift and offset, all isotope standards are measured every 25 hours. The 25-hour cycles are chosen to avoid missing the same time of day for all daily observations due to the calibration routine. The measured isotope values are compared to the expected real standard isotope values for offset correction and measured isotope values over a period of 14 days are considered for drift correction. Screening of the data for special events with anomalous vapour or isotope data are performed afterwards.

During the calibration procedure, water vapour is produced from the liquid isotope standard within the bubbler system. Over the course of a year, this might lead to a change in the isotopic composition of the liquid standards. To correct for a potential

change in the standards, samples from all liquid standards are taken and measured yearly. For this study, the last sampling was at the end of the campaign in February 2019.

The uncertainty of measurements contains the accuracy and the precision on corrected measurements using error propagation method. The accuracy is calculated based on a calibration program, considering the instrumental drift, deviation from known isotopic values, and systematic error according to humidity response functions and the precision is based on taking average on measured data for 1-hour corrected data as the output of the calibration program. From the determined humidity response functions and calibration procedure, we estimate the mean uncertainty of the Picarro isotope data over the whole observational period as 0.45 ‰ for $\delta^{18}O$, 2.99 ‰ for $\delta D$, and 3.03 ‰ for $d$ values.

### 2.4 Moisture source diagnostics

To study the origin and transport paths of water vapour to Neumayer Station, the Lagrangian particle dispersion model FLEX-PART (Brioude et al., 2013) enhanced by a Lagrangian moisture source diagnostic (Sodemann et al., 2008) is used in this study. Meteorological data needed for the FLEXPART model are taken from the ERA5 reanalysis dataset (Hersbach et al., 2020), provided by the European Centre for Medium-Range Weather Forecasts (ECMWF). To retrieve and prepare ECMWF data for use in FLEXPART, the software package Flex_extract v7.1.2 has been used (Tipka et al., 2020).

Air parcels are traced backwards from the final destination (Neumayer Station) for 10 days, similar to the setup described by Sodemann et al. (2008). The moisture source diagnostic based on the Lagrangian back-trajectories provides values of "moisture uptake" (in $mm\ day^{-1}$) on a $1° \times 1°$ grid. This parameter represents the amount of moisture injected to the air masses within each grid cell, contributing to the humidity at Neumayer Station.

## 3 Results

### 3.1 Temperature and water vapour measurements

For the observational period 17 February 2017 until 22 January 2019, daily mean values of temperature, specific humidity, relative humidity, $\delta^{18}O$, $\delta D$, and $d$ have been determined (Fig. 2). In addition, multi-year daily mean values of temperature, specific humidity and relative humidity over the 38-year period from 1981-2018 have been calculated. Meteorological data (temperature, relative humidity, and specific humidity) are weather station data measured in a distance of 50 meters from the main station building at a height of 2 meters, and the isotope values have been measured with the Picarro instrument, sampling air from the station roof at a height of 24 meters.

There exist some data gaps for isotope values for these two years of measurements. Water vapour isotope data is missing at some days because of maintenance or reparation of the instrument, measuring humidity response functions, or due to the removal of data outliers related to instable measurements of the Picarro instrument. In total, daily vapour and isotope data exists for 600 out of 705 days (85 %).

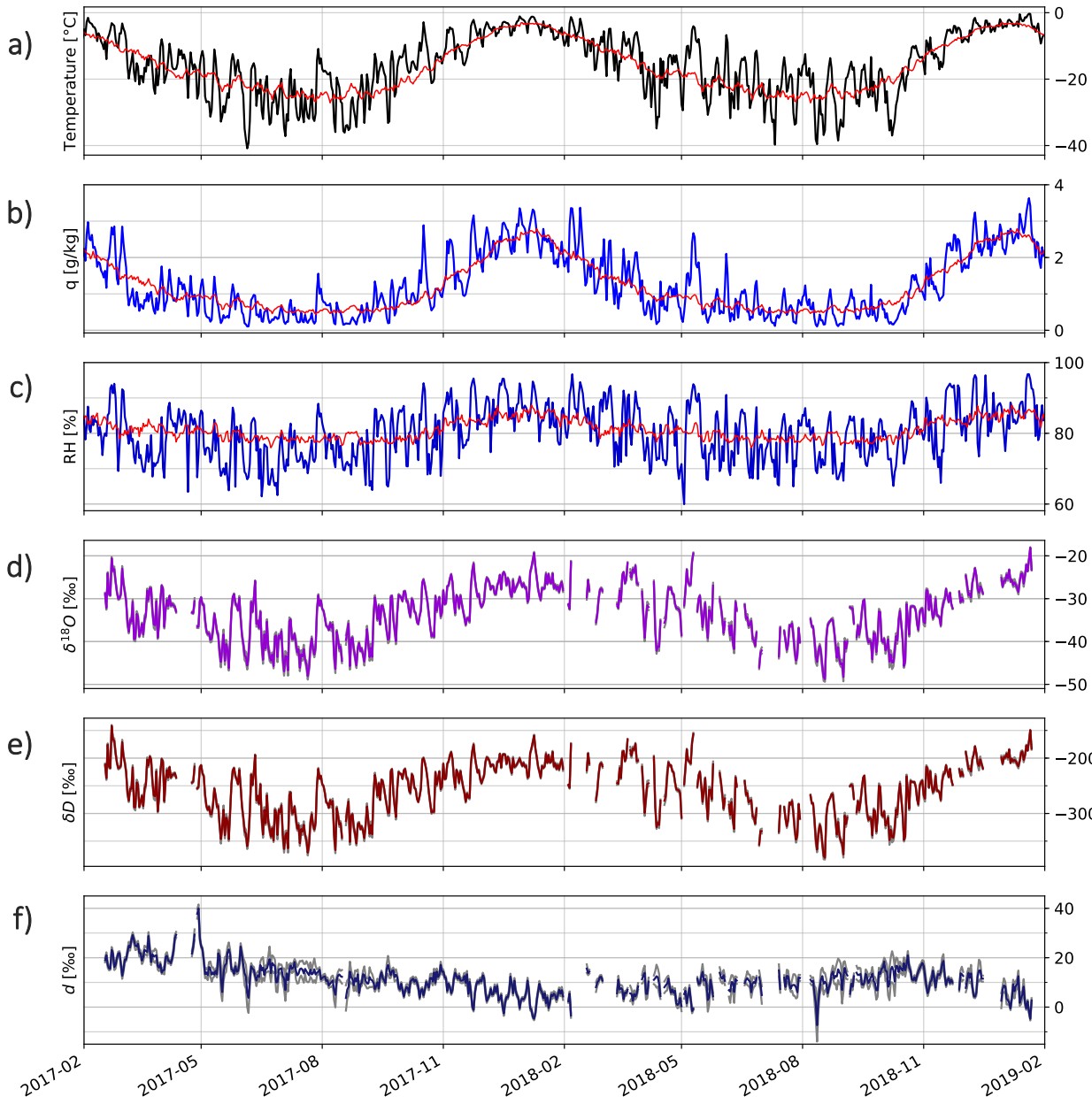

**Figure 2.** Daily averaged observations at Neumayer Station from February 2017 to January 2019. Downward: a) 2-m temperature [$^\circ$C]; b) specific humidity [ $g\ kg^{-1}$]; c) relative humidity [%]; d) $\delta^{18}O$ [‰]; e) $\delta D$ [‰]; f) $d$ [‰]. To have a better comparison, the climatology (multi-year daily average temperature, specific humidity, and relative humidity over the 38-year period from 1981 to 2018) is shown with a red line in meteorological observations. The determined uncertainties of the Picarro instrumental data (see text) are plotted as gray lines.

We compare the specific humidity measured by the Picarro instrument with the specific humidity values measured routinely as part of the meteorological observations at Neumayer Station (Schmithüsen et al., 2019). The relationship between these two series of humidity measurements is $q_{(Picarro)} = 1.5 q_{(meteorology)} + 0.08$ ($N = 12198$, hourly values between 17 February 2017 and 22 January 2019, $r = 0.97$, standard error of the estimate = 0.0022 $g$ $kg^{-1}$; Fig. 3). The rather high slope between both humidity measurements and also a number of unusual high and low Picarro humidity values motivated us to analyse the difference between both humidity data sets in more detail.

The inlet of the Picarro instrument is situated approx. 17.5 m above the surface level of the station. As the station is placed on a small artificial hill, this surface level is approx. 7.6 m higher than the surface level of the meteorological mast placed 50 meters beside the station building. Thus, the total height difference between the Picarro inlet and the height of the meteorological humidity measurements is approx. 22 m. In principle, higher humidity values at the Picarro inlet could be explained by a humidity inversion layer above the surface, which could remove near-surface moisture at the meteorological mast position by hoar frost formation. However, temperature differences between a 2-meter temperature sensor at the meteorological mast and temperatures measured on the station's roof do not exceed 2 $°C$ during our measurement period. No strong temperature inversions are found for the days with extreme Picarro humidity measurements.

To test if contamination by exhaust gases could be another reason for the data mismatch, the wind direction was analysed for those hourly Picarro humidity values which are much higher than the corresponding humidity values measured by the meteorological station. Most of the outliers coincide with a wind direction from the south (and a few from the east), while the Picarro inlet has been placed east of the exhaust gases origin. Thus, the possibility that contamination by exhaust gases is the reason for the unusually high Picarro humidity values is excluded.

Picarro humidity measurements have been compared with independent humidity observations for a few studies; so far. Aemisegger et al. (2012) calibrated and controlled the humidity of their Picarro instrument by a dew point generator and showed a linear relationship between Picarro measurements and the humidity measured by the calibration system with a slope of 1.27 and an uncertainty of $\sim$100–400 ppm (0.06-0.24 $g$ $kg^{-1}$). Tremoy et al. (2011) reported that the slope between humidity measured by a meteorological sensor and humidity measured by a Picarro instrument can change from 0.81 to 1.47 depending on site conditions. Bonne et al. (2014) also showed a non-linear response of their Picarro instrument compared with the humidity measured by a meteorological sensor. Based on their data, the ratio between Picarro humidity and sensor humidity values changed from 1 to 1.87, depending on the amount of humidity. Compared to the results of these studies, we rate the calculated ratio of our Picarro humidity measurements versus the humidity data from the meteorological mast ($q_{(Picarro)}/q_{(meteorology)} = 1.5$) as unobstructive.

As in previous studies (e.g., Bonne et al., 2014) we will use the Picarro humidity data to calculate of the humidity response functions required for the calibration of the isotope measurements. All analyses regarding the relationships between water vapour isotopes and local climate variables, on the other hand, are based on the humidity and corresponding temperature data measured at the meteorological mast.

Daily temperatures at Neumayer Station vary between approx. 0°C in austral summer and -40°C in austral winter. Daily values of the specific humidity vary between 0.06 $g$ $kg^{-1}$ (corresponding to water concentration approx. 100 ppm) and 3.75

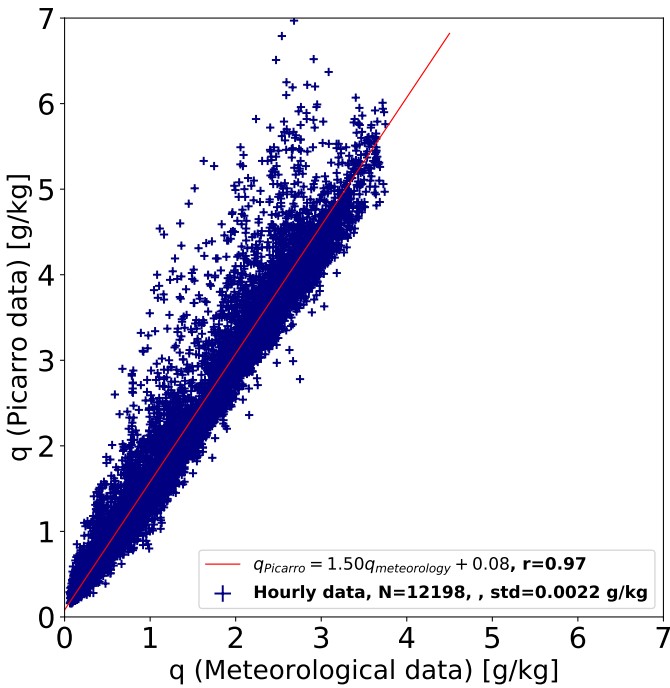

**Figure 3.** Hourly averaged specific humidity values measured by a meteorological sensor vs. specific humidity values measured by the Picarro instrument at Neumayer Station from February 2017 to January 2019. Specific humidity values are given in $g\ kg^{-1}$.

$g\ kg^{-1}$ (approx. 6000 ppm). Daily relative humidity fluctuates between 59.95 % and 96.71 % with a mean value of 80.42 %. Daily $\delta^{18}O$ ($\delta D$) values of the vapour vary in a range between -48.79 ‰ and -18.10 ‰ (-380.07 ‰ and -141.67 ‰). For $d$, the values range approx. from -5 ‰ to +35 ‰. The annual mean values of all variables except $d$ do not show a significant change (the change is less than the mean uncertainty for hourly data) between the first and second year of measurements. The average 2-m temperature for the year 2017 and 2018 is -15.81°C and -15.65°C, respectively. Both values are close to the long-term annual temperature of -16.1°C (mean of the years 1981 to 2018). For $\delta^{18}O$ ($\delta D$), the mean annual values for the first and second year of measurements are -33.30 ‰ and -33.08 ‰ (-253.35 ‰ and -255.17 ‰), respectively. For $d$, mean annual values for the first and second year are 13.06 ‰ and 9.50 ‰. The annual average of the specific humidity (relative humidity) for 2017 and 2018 is 1.18 $g\ kg^{-1}$ (79.75 %) and 1.23 $g\ kg^{-1}$ (80.60 %).

Temperature, specific humidity, relative humidity, $\delta^{18}O$, and $\delta D$ show a clear seasonal cycle in the daily mean data series (Fig. 2) with high values at the end of austral summer (January–February) and low values at the end of austral winter (August–September). No such seasonal cycle is observed for the $d$ values. While $d$ values are constantly decreasing between February 2017 and February 2018, small variations with a peak in autumn 2018 can be detected until January 2019. Due to the limited period of two years, the dataset is too short to determine if any seasonal cycle of $d$ exists in our measurements. This is even

more true, as the uncertainty of the measured isotope values depends on the specific humidity amount, which is much lower at Neumayer Station in austral winter as compared to the summer season. Based on the determined mean uncertainty of the 1-hour Picarro data (Section 2), we estimate the monthly average uncertainty for $d$ in austral winter as high as 4.5 ‰ while it decreases during austral summer down to 1.9 ‰ (see Appendix, Table B1).

## 3.2 Relationships between water vapour isotopes and local climate variables

We analyse the relationship between $\delta^{18}O$, $\delta D$, temperature, specific humidity, and relative humidity to determine the key meteorological variables controlling the isotope signals in vapour at Neumayer Station.

### 3.2.1 $\delta^{18}O$ vs. temperature and humidity

$\delta^{18}O$ (and also $\delta D$) is strongly correlated with the 2-m temperature. For daily averaged values, the relationship between 2-m temperature and $\delta^{18}O$ in water vapour is $\delta^{18}O = 0.58T - 24.01$ ($r = 0.89$).

Next, we look at different seasons of the year and evaluate the slope and correlation coefficient of the $\delta^{18}O - T_{2m}$-relationship for each season, considering daily average values. The slopes for spring (SON), summer (DJF), autumn (MAM), and winter (JJA) are $0.58 \pm 0.03$ ‰ $°C^{-1}$ ($r = 0.86$), $0.68 \pm 0.06$ ‰ $°C^{-1}$ ($r = 0.71$), $0.63 \pm 0.04$ ‰ $°C^{-1}$ ($r = 0.83$), and $0.48 \pm 0.03$ ‰ $°C^{-1}$ ($r = 0.75$), respectively (Fig. 4).

To look at the effect of temperature on isotopic compositions in term of air-masses and large-scale circulation, we examine a higher-level temperature (850 hPa), using ECMWF, ERA5 dataset (Hersbach et al., 2020). The average 850 hPa temperature for the year of 2017 and 2018 is -14.45°C, which is about 1°C warmer than the observed 2-meter average temperature. Daily values of the 850 hPa temperature vary between -31.09°C and -2.97°C, showing a smaller amplitude compared to the 2-meter temperature values at Neumayer Station. For the observational period, the correlation coefficient between the 850 hPa temperature and $\delta^{18}O$ ($r = 0.68$) is less than the one between the 2-meter temperature and $\delta^{18}O$ ($r = 0.89$). We can see this characteristic also in a seasonal view. The correlation coefficient for the 850 hPa temperature and $\delta^{18}O$ (the observed 2-meter temperature and $\delta^{18}O$) for different seasons is calculated: spring: $r = 0.60$ ($r = 0.86$); summer: $r = 0.21$ ($r = 0.71$); autumn: $r = 0.52$ ($r = 0.83$); and winter: $r = 0.43$ ($r = 0.75$).

Daily temperature and $\delta^{18}O$ values in summer are less fluctuating than in the other three seasons (Fig. 4). This might be explained by a weaker temperature inversion, lower sea ice variability, and stronger sublimation and snowmelt in summer.

Hudson and Brandt (2005) showed that the temperature inversion strength variations in winter are one reason for the large day-to-day variability of 2-meter temperature in Antarctica. In winter, clouds can be much warmer than the surface, which leads to a strong temperature inversion. However, changes in wind speed and direction might change the cloud cover and thereby weaken or destroy the inversion layer, in short time. Due to these processes, stronger temperature inversions can lead to higher temperature variability in winter.

As sea ice can strongly limit the heat flux between a relatively warm ocean and the atmosphere, sea ice coverage variations close to Antarctica's coastal stations can primarily affect the near-surface temperature at the stations (Turner et al., 2020).

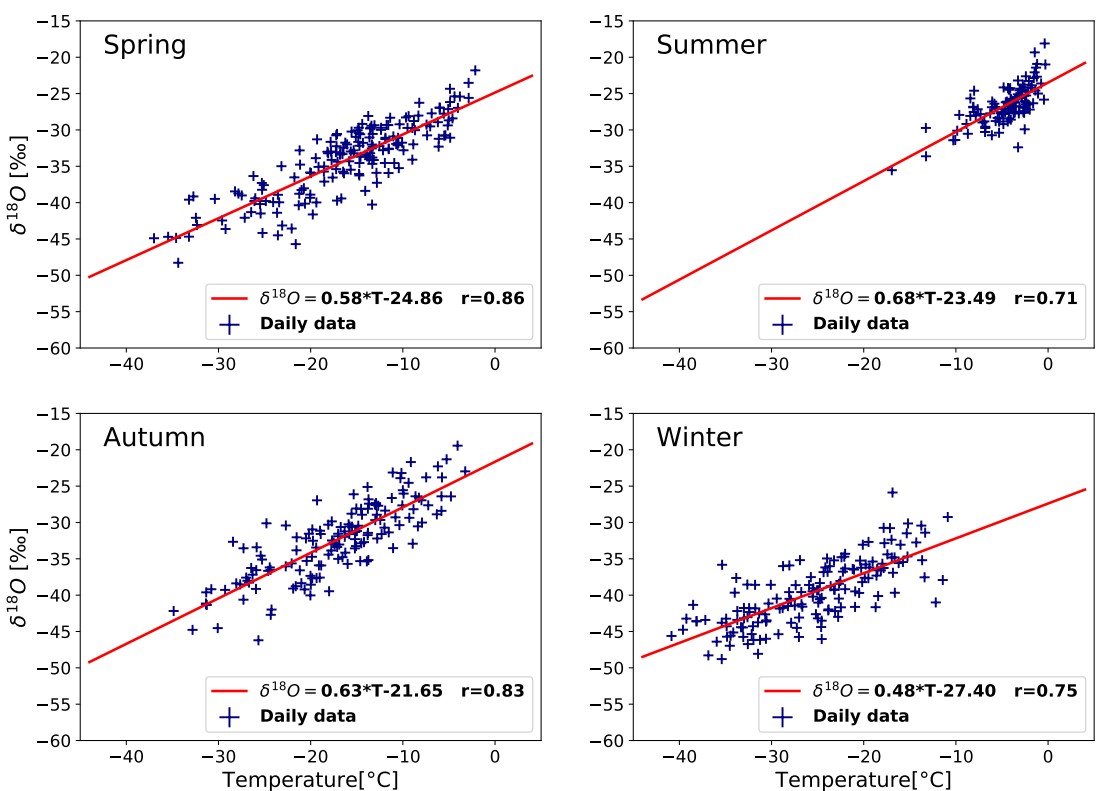

**Figure 4.** $\delta^{18}O$ [‰] vs. temperature [°C]. Four plots show daily average temperature–$\delta^{18}O$ values for different seasons of the year. For each season, a best fitted line, using the least-squares approach, for $\delta^{18}O$ vs. temperature is plotted as a red line and corresponding correlation coefficients are calculated.

Decreasing sea ice variability close to the Neumayer Station in summer compared to other seasons, which is true for most other coastal stations in Antarctica, may also lower the temperature variability.

Another reason for the reduced temperature variations in summer can be a stronger heat loss, which prevents temperatures above zero. At Neumayer Station, the largest sources of heat loss in summer are sublimation and snow melting (Jakobs et al., 2019). The sublimation is primarily temperature-controlled and is only significant at Neumayer station in summer. About 19% of the annual snowfall at this location is removed by sublimation (van den Broeke et al., 2010). The second source of heat loss at the station is snow melting. In summertime, when the air temperature can rise above $0°C$, the surface snow will reach its melting point and start to melt. For the melting process, the incoming radiative energy is partly used for latent heat uptake, keeping the near-surface temperature close to the melting point.

These three phenomena might explain the detected cut-off at 0°C of the 2-m temperature (Fig. 4). They could also partly explain the lower correlation coefficient between the 2-m temperature and $\delta^{18}O$ in summer, as upper air temperatures most likely control the latter.

At Neumayer Station, the specific humidity is highly correlated with temperature (Jakobs et al., 2019), as expected from the general Clausius-Clapeyron relation between both quantities. As a consequence, the $\delta^{18}O$ values of water vapour at Neumayer Station are strongly correlated not only to temperature, but also to specific humidity ($r = 0.85$).

As isotopic fractionation is primarily controlled by temperature, for the typical synoptic situation at Neumayer Station, we estimate temperature fluctuations as the main driver for both changes in specific humidity and the $\delta^{18}O$ signal in water vapour. The correlation for $\delta^{18}O$ and specific humidity also varies in different seasons following the correlation coefficient between temperature and $\delta^{18}O$. The correlation coefficient for relative humidity and $\delta^{18}O$ in different seasons are similar (spring: $r = 0.73$, summer: $r = 0.57$, autumn: $r = 0.69$, and winter: $r = 0.65$).

### 3.2.2 $\delta^{18}O$ vs. $\delta D$ and $d$

Daily values of $\delta^{18}O$ and $\delta D$ reveal a high correlation coefficient ($r = 0.99$), with a $\delta^{18}O$–$\delta D$ slope of $7.67\,\text{\textperthousand}\,\text{\textperthousand}^{-1}$ (Appendix, Fig. B1). For the Deuterium excess, an overall weak negative correlation between $\delta^{18}O$ and $d$ ($r = -0.35$) is found. The anti-correlation between $\delta D$ and $d$ is weaker ($r = -0.24$). For $d$, the correlation with 2-m temperature is low and negative ($r = -0.33$). The corresponding correlation of the specific humidity with $d$ is stronger ($r = -0.48$).

Analysing different seasons of the year, a negative correlation between $\delta^{18}O$ and $d$ is detected for spring, $r = -0.47$, summer, $r = -0.51$, and autumn, $r = -0.37$, but for winter, no correlation exists. This pattern can be detected also for temperature-$d$, specific humidity-$d$, and relative humidity-$d$ relations.There is a negative correlation coefficient between temperature and $d$ for spring, $r = -0.41$, summer, $r = -0.60$, and autumn, $r = -0.14$, but in winter a weak positive correlation, $r = 0.22$, is noticed. There are anti-correlations between the specific humidity (relative humidity) values and $d$ for spring, $r = -0.50$ ($r = -0.43$), summer, $r = -0.71$ ($r = -0.59$), and autumn, $r = -0.24$ ($r = -0.19$), which are slightly stronger than the ones between temperature and $d$. For winter, there is a weak positive correlation between the specific humidity (relative humidity) and $d$, $r = 0.13$, ($r = 0.04$).

### 3.3 Moisture source uptake and vapour transport

For further understanding of the seasonal different relations between $\delta^{18}O$, $\delta D$, Deuterium excess, and the meteorological variables, we analyse potential seasonal differences in the main moisture uptake areas for vapour transported to Neumayer Station for years of 2017 and 2018, as simulated by FLEXPART (Fig. 5). Moisture uptake coming to Neumayer Station depends on different factors such as sea ice extent, the Southern Hemisphere semi-annual oscillation (SAO), and absolute temperature. As Fig. 5 shows, the sea ice prevents evaporation from the ocean. In the areas with ice coverage more than 90 %, the moisture uptake is minor. The SAO is the main phenomenon that affects surface pressure changes at the middle and high latitudes of the Southern Hemisphere (Schwerdtfeger, 1967). It means the twice-yearly contraction and compression of the pressure belt surrounding Antarctica as a result of the different heat capacities of the Antarctic continent and the ocean. The

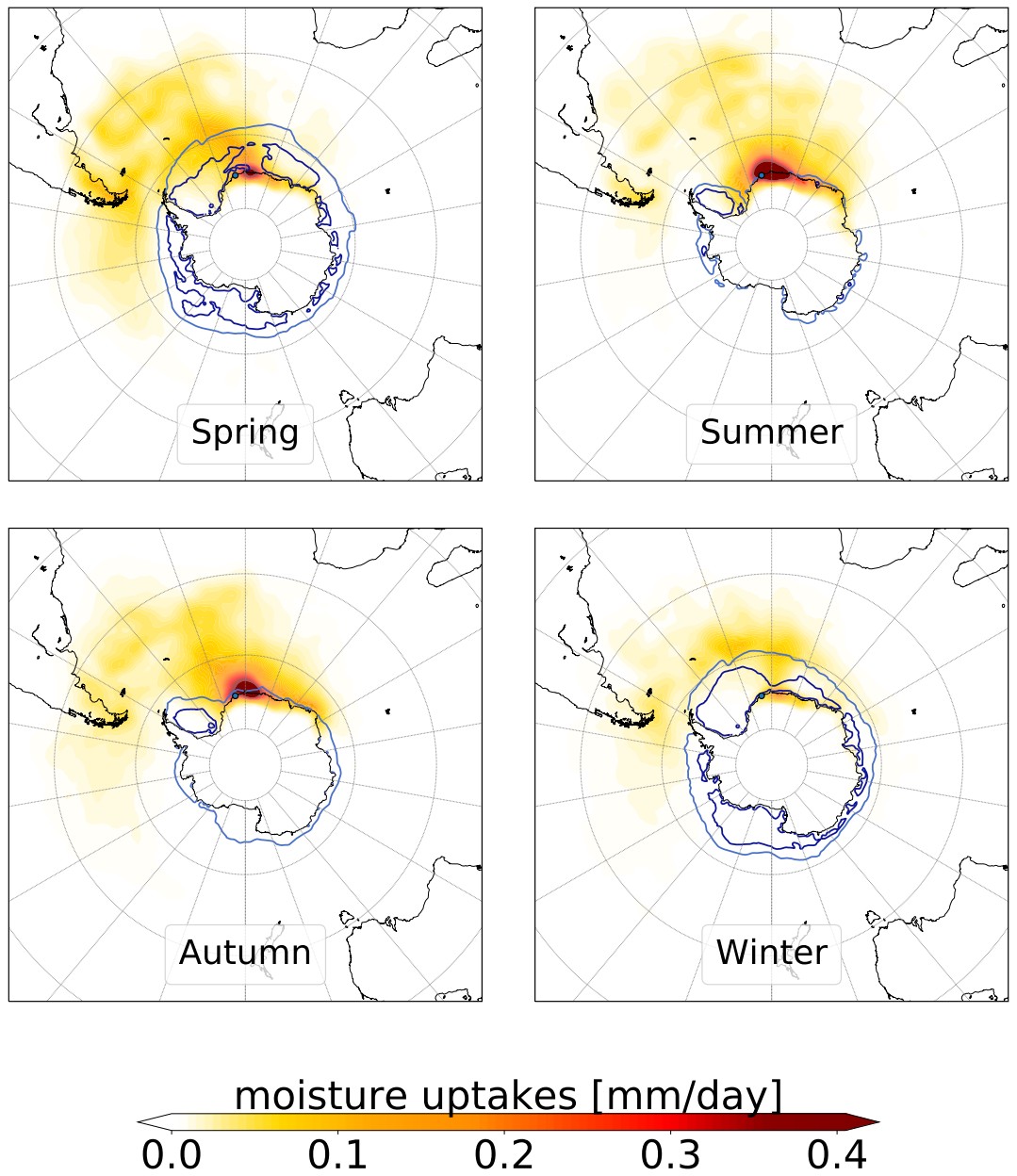

**Figure 5.** Simulated mean moisture uptake occurring within the boundary layer $[mm\ day^{-1}]$ in the pathway to Neumayer Station during last 10 days modelled by FLEXPART using ECMWF, ERA5 dataset (Hersbach et al., 2020), for spring (SON), summer (DJF), autumn (MMA), and winter (JJA), considering the year of 2017 and 2018. The mean sea ice edge based on ERA5 reanalysis (Hersbach et al., 2020) for ice coverage more than 45% and 90% is shown with light blue and dark blue lines.

SAO leads to a clear half-yearly pressure wave in surface pressure at high latitudes and modifies the atmospheric circulation and temperature cycles (Van Den Broeke, 1998).

In spring, major moisture uptake and transport happens from oceanic areas north-west of the station at high to mid-latitudes. In summer, most of the moisture uptake occurs in the coastal areas close to the station and in the South Atlantic Ocean, although in summer more humidity comes to the station. In autumn, moisture uptake occurs again mainly close to or east of the station, similar to summer. For winter, the moisture amount transported to the station is substantially less than in other seasons and comes from a wide area of the Southern Ocean, partly even from the Pacific.

## 3.4 Wind and pressure pattern

The origin of the air masses measured at Neumayer Station depends directly on the local wind, which is characterized by relatively high wind speeds, with an annual mean value of $8.7\ m\ s^{-1}$ during the measurement period (with a standard deviation of $5.67\ m\ s^{-1}$, considering daily values of all days). Two main wind directions are observed (Fig. 6a, left). The prevailing wind direction is from east, caused by the passage of cyclones north of the Antarctic coast in the circumpolar trough (Fig. 7a, as an example of a day with wind from east) and the second, less frequent, typical wind direction is south to south-west, caused by a mixture of weak katabatic and synoptic influence, when Neumayer Station is situated between a cyclone to the east and an anticyclone to the west (Fig. 7b, as an example of a day with a main wind from south and south-west) (König-Langlo et al., 1998; König-Langlo and Loose, 2007; Rimbu et al., 2014). Pure katabatic winds are rare and restricted to extended high-pressure systems. Because the Ekström ice shelf slope is gently upward to the south, katabatic wind remains below 10 $m\ s^{-1}$ (south-north direction).

Wind direction and wind speed are two of the main factors used in our back trajectory calculations to determine the origin of air parcels heading to Neumayer Station. To check the robustness of our results, we compare ERA5 data wind data, which is used in this study for FLEXPART simulations to identify moisture sources and transport pathways to Neumayer Station, with meteorological observations from Neumayer Station. The comparison reveals that the ERA5 dataset reproduces wind direction and wind speed around Neumayer Station well for most days (Fig.6). However, for cold events (as defined in Section 2.2), the ERA5 dataset has a bias with respect to the katabatic winds (Fig. 6c), although the Neumayer Station meteorological data is assimilated in ERA5. The main reason for this bias in the ERA5 data might be the low number of stations in Antarctica, which are used to generate the ERA5 reanalysis dataset. Due to this bias in the ERA5 data, the simulated moisture uptake and vapour transport pathways during cold events, when katabatic winds from the south and south-west occur at Neumayer Station, should be taken with caution.

In addition to the local wind at Neumayer Station, vapour transport to the station is also strongly controlled by the larger-scale pressure and wind pattern. For all seasons, generally there is a high-pressure system above the Antarctic continent and a low-pressure belt surrounding the coast, thus, north of the station. This pressure pattern puts the station on average at the southern edge of a low-pressure system, which leads to a cyclonic circulation that transports vapour from the Southern Atlantic to Neumayer Station from easterly directions. As the pressure pattern related to this circulation is weakened in summer, far-field transport of vapour to Neumayer Station is reduced as compared to other seasons.

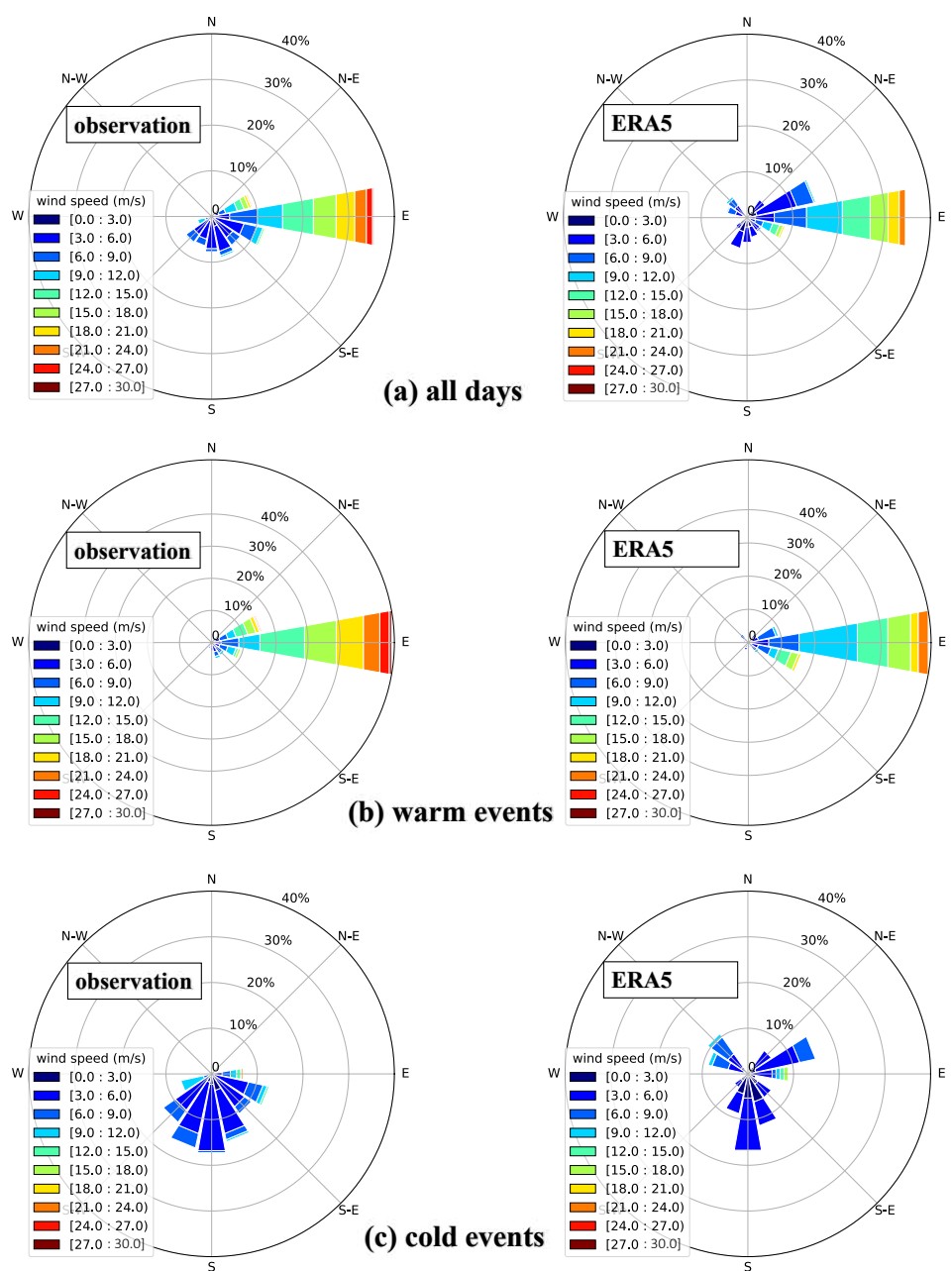

**Figure 6.** Two-dimensional frequency daily 10-m wind (stated in $m\ s^{-1}$) at Neumayer Station during the year of 2017 and 2018 provided by observations and ECMWF (ERA5) data for (a) all days of the year, (b) days considered as warm events, (c) days considered as cold events.

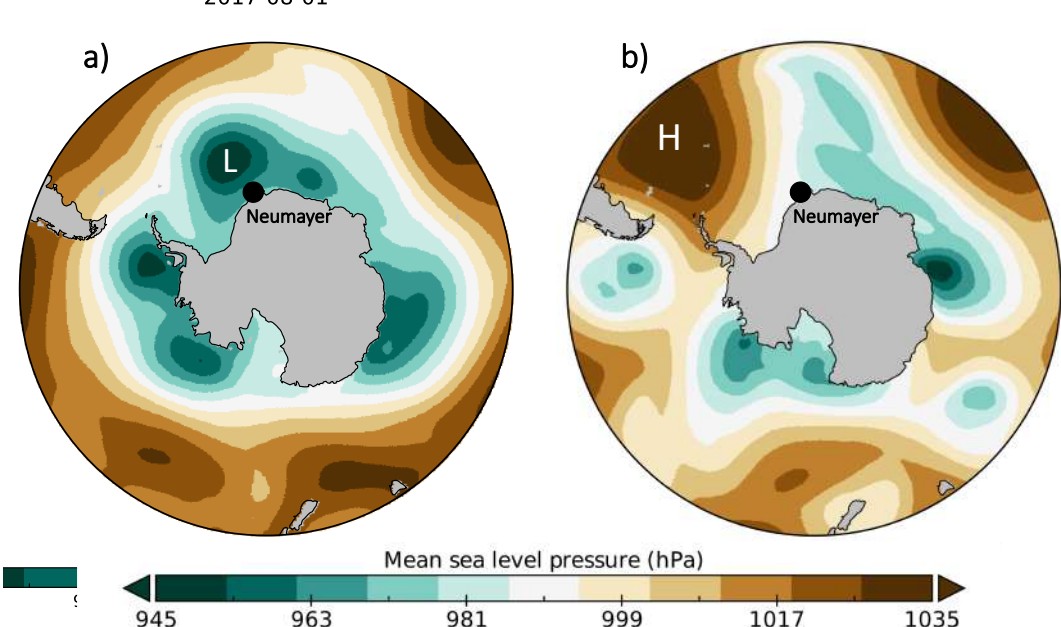

**Figure 7.** Mean sea level pressure for two different days: a) 2017-08-01, with a main wind from east; b) 2017-08-09, with a main wind from south (katabatic wind). The low pressure system close to the station is marked by "L" and the high pressure one is marked by "H". Pressures are given in [hPa]. Data are from ERA5 data set, ECMWF (Hersbach et al., 2020).

### 3.5 Diurnal cycle

To evaluate the diurnal cycles at Neumayer Station, we consider two months of two sequent summers (December-January of 2017/18 and 2018/19) in order to compare our results with previous studies performed in Antarctica. We derive the daily mean values (the mean of 24 hourly mean values for each day) and subtract it from the time series. The remaining anomalies of all parameters represent an average diurnal cycle (Fig. 8).

The average of all values of each variable for the diurnal cycle study period (December-January of 2017/18 and 2018/19) are: $\delta^{18}O$: $-26.34\,\permil$; $\delta D$: $-205.27\,\permil$; $d$: $5.46\,\permil$; 2-meter temperature: $-4.25\,°C$; 10-meter temperature: $-3.87\,°C$; specific humidity: $2.49\ g\ kg^{-1}$; relative humidity: $86.30\ \%$; wind speed: $7.53\ m\ s^{-1}$; wind direction: $291\ degree$ (we consider only winds with a wind speed of more than $3\ m\ s^{-1}$); shortwave downward radiation: $228.98\ W\ m^{-2}$.

Strong diurnal cycles in 2-meter temperature (Fig. 8d, red line), 10-meter temperature (Fig. 8d, green line), specific humidity (Fig. 8e), and relative humidity (Fig. 8f) are detected. For wind speed, the diurnal cycle is weak (Fig. 8g) and for wind direction no diurnal cycle is detectable (Fig. 8h). In summer, there is no strong temperature inversion close to the surface, at least not for the first 10 meters above surface. The temperature differences between 2-meter and 10-meter height reaches up to $1\ °C$ during the coldest time of a day, while during half of the day their difference is less than $0.4\ °C$.

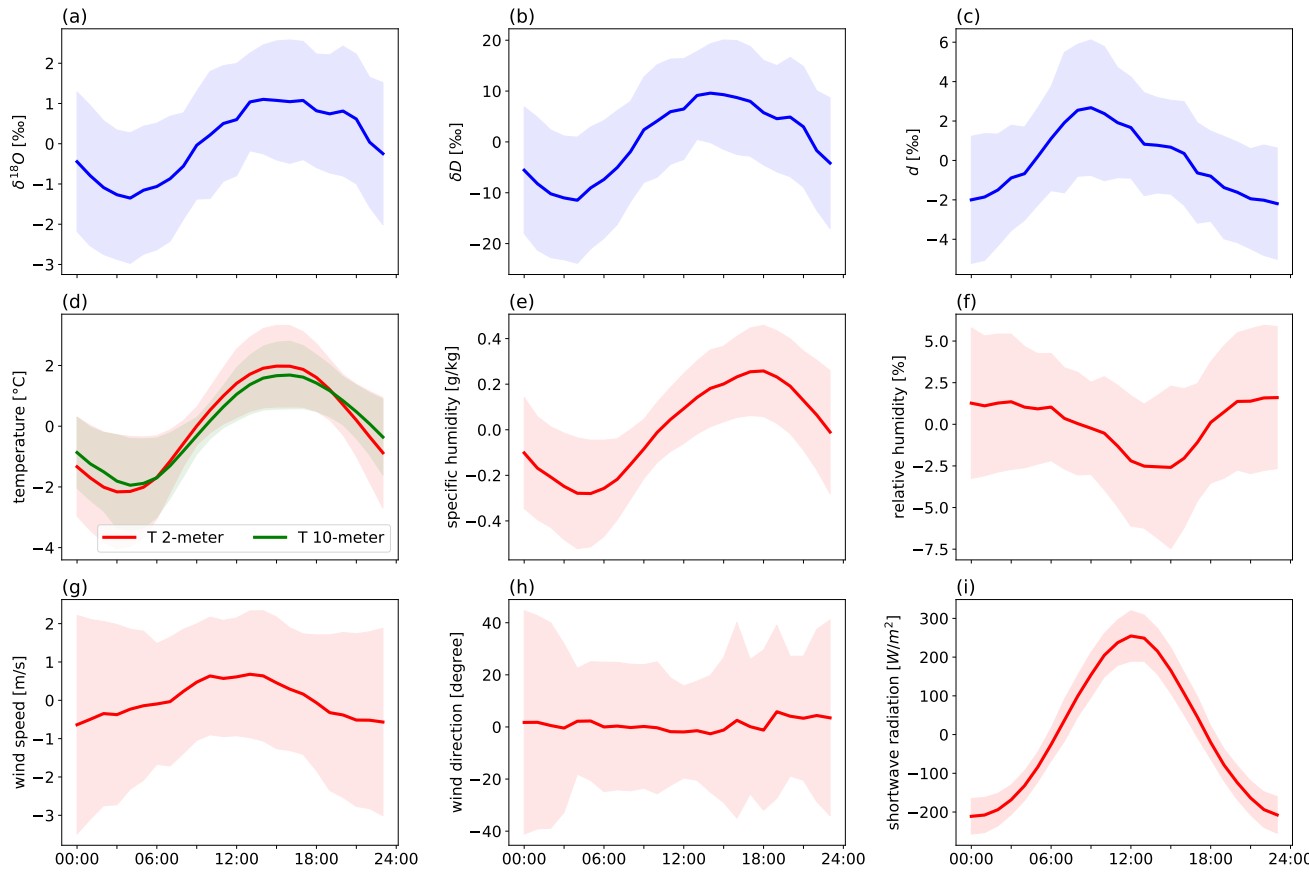

**Figure 8.** Anomaly diurnal cycles of (a) $\delta^{18}O$ [‰], (b) $\delta D$ [‰], (c) $d$ [‰], (d) 2-meter temperature and 10-meter temperature [$^\circ C$], (e) specific humidity [$g\ kg^{-1}$], (f) relative humidity.[%], (g) wind speed [$m\ s^{-1}$], (h) wind direction [degree], (i) shortwave downward radiation [$W\ m^{-2}$], and their $\pm 1$ standard deviations for two months of two sequent summers (December-January of 2017/18 and 2018/19). Blue colour shows the Picarro instrument measurements and red colour (and green) shows meteorological observations at Neumayer Station. The local time zone at Neumayer Station is equal to UTC time.

The amplitudes of 10-meter temperature (3.63 $^\circ C$) are less than 2-meter temperature (4.14 $^\circ C$). The amplitudes of the specific humidity and relative humidity are 0.54 $g\ kg^{-1}$ and 4.19 %, respectively.

A clear diurnal cycle can be detected for $\delta^{18}O$ (Fig. 8a), $\delta D$ (Fig. 8b), and $d$ (Fig. 8c). The diurnal amplitudes are 2.45 ‰, 21.07 ‰, and 4.87 ‰, respectively. A very high correlation coefficient between $\delta^{18}O$ and 2-meter temperature ($r = 0.98$) and 10-meter temperature ($r = 0.99$) suggests the temperature changes as the main driver of water vapour $\delta^{18}O$ diurnal variations. $d$ is rather anti-correlated with relative humidity ($r = -0.59$), while it does not show a considerable correlation with temperature and specific humidity.

The 2 and 10-meter temperature cycles and consequently $\delta^{18}O$ and $\delta D$ follow the shortwave radiation (Fig. 8i) with a short delay (around 3 hours) and show the minimum and maximum values at 03:00 UTC (local time) and 15:00 UTC. The relative humidity behaves the other way round and shows the minimum value at 15:00 UTC and and maximum values between 21:00 UTC and 03:00 UTC. $d$ has the minimum at 00:00 UTC and the maximum at 09:00 UTC.

## 4 Discussion

### 4.1 Key controls on vapour $\delta^{18}O$

#### 4.1.1 Temperature and humidity

Relatively high correlations between daily $\delta^{18}O$ and temperature ($r = 0.89$), and also between daily $\delta^{18}O$ and specific humidity ($r = 0.85$), highlight the role of temperature and humidity as the main drivers of $\delta^{18}O$ variations in water vapour. When the temperature of an air parcel decreases, water vapour condenses, which results in a decreasing $\delta^{18}O$ value in water vapour and also decreasing humidity in the air parcel. Neumayer Station is situated at the southern edge of a low-pressure system (explained in Section 2.2) and the main air path is from east caused by a cyclonic circulation, which is clockwise in the Southern Hemisphere. This air path transports vapour from the ocean (lower latitude) to the station (higher latitude). Due to the temperature decrease, the specific humidity and as a result the $\delta^{18}O$ value drop.

#### 4.1.2 Wind

Next, we analyse the impact of another meteorological factor, wind, on the isotope values in vapour at Neumayer Station. During the observation period, on 86 % of all days that involve warm events at Neumayer Station, the wind came from east. Such wind conditions are usually a result of a low-pressure system north of the station (König-Langlo et al., 1998; König-Langlo, 2017). In such a situation, the weather at Neumayer Station is typically relatively warm with high relative humidity (88% of days with a relative humidity higher than 90% coincide with wind from east) and cloudiness (85% all cloudy days, i.e. days with a total cloud amount of more than 80%, coincide with this wind direction). Relatively higher temperature and specific humidity lead to more enriched $\delta^{18}O$ values. During days that involve cold events, the winds typically come from south to south-west and wind speeds are rather low. This weather pattern occurs when a cyclone has moved eastward, so that the former low-pressure area is replaced by a high-pressure ridge. In such a situation, wind speeds decrease and the wind direction changes from easterly to southerly and south-westerly. The weak katabatic winds are strengthened by the synoptically caused air flow and bring cold and dry air from East Antarctic Plateau to Neumayer Station, usually dissolving the clouds. Lower temperature and specific humidity result in more depleted $\delta^{18}O$ in water vapour coming to Neumayer Station. General wind patterns and their effects at Neumayer Station are sketched in Fig. 10.

In order to filter out the dominant temperature control on $\delta^{18}O$ in this analysis, we proceed as follows: we calculate a "predicted" daily $\delta^{18}O$ value in vapour, based on the corresponding daily temperature value and the observed relationship between temperature and $\delta^{18}O$ in vapour ($\delta^{18}O = 0.58T - 24.01$; see above). Then we look at the residual between the predicted $\delta^{18}O$

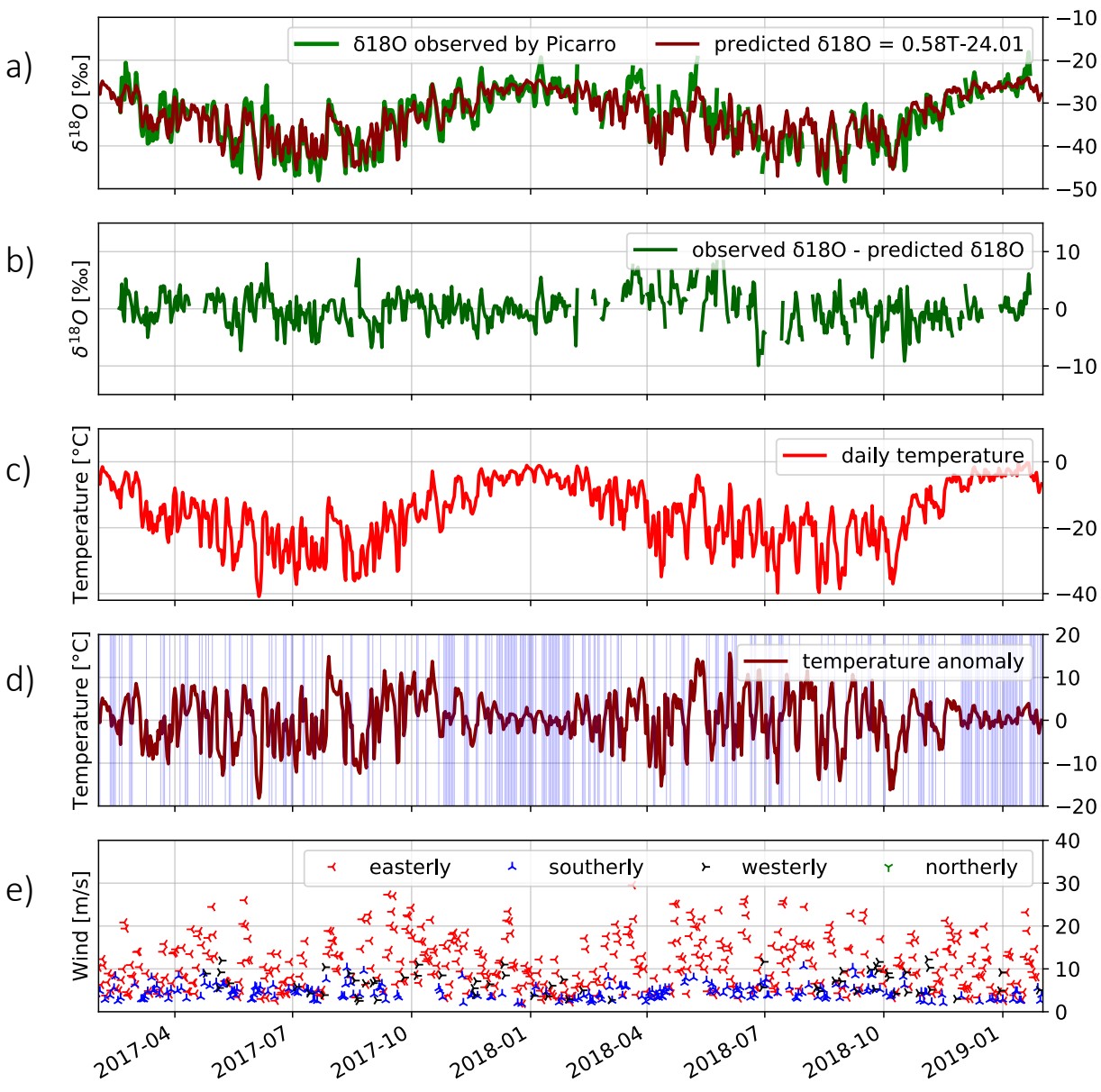

**Figure 9.** Daily observation time series at Neumayer Station from February 2017 to January 2019. Downward: a) green: observed $\delta^{18}O$, dark red: predicted $\delta^{18}O$ based on the annual relation between temperature and $\delta^{18}O$; b) observed $\delta^{18}O$ minus predicted $\delta^{18}O$; c) 2-m temperature [°C]; d) 2-m temperature minus climatology for 2-m temperature (multi-year daily 2-meter temperature averages from 1981 to 2018). Days with no temperature events are highlighted with blue colour; e) wind speed [ $m\ s^{-1}$ ] and wind direction: red: easterly, blue: southerly, black: westerly, and green: northerly.

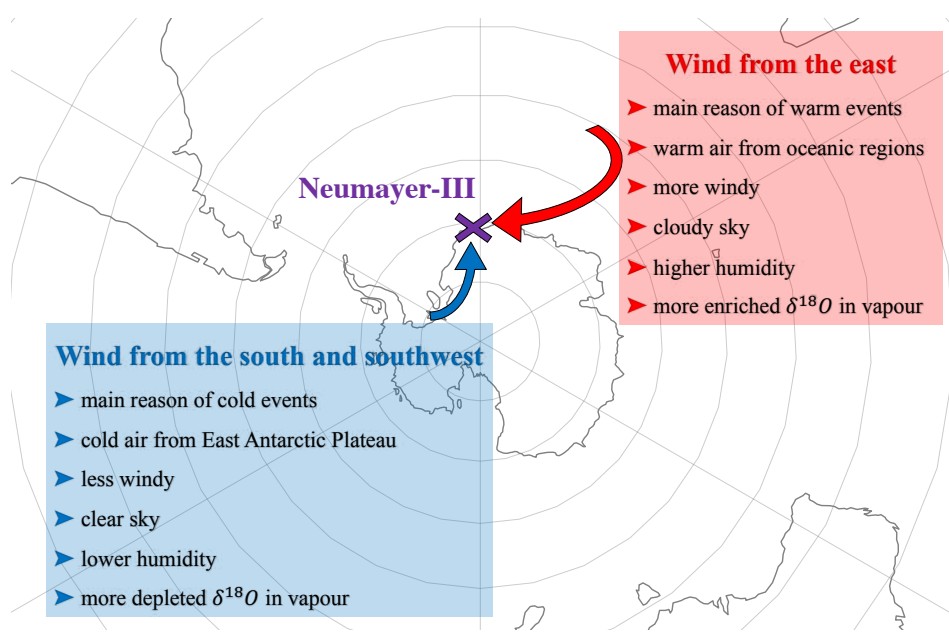

**Figure 10.** Frequent wind patterns (the first, easterly wind, and the second, southerly and south-westerly wind) at Neumayer Station and their characteristics.

value and the observed $\delta^{18}O$ value (Fig. 9) and analyse how this residual might be linked to different wind pattern at Neumayer Station. Here we include in our analyses only days with no warm and cold events (as defined in Section 2.2). We select the days without significant positive or negative deviations from the long-term mean temperature and check if their $\delta^{18}O$ values are unusually high or low, and also determine the wind speed and direction of these days. We find that for 64 % of the days (22 out of 36 days) with the wind coming from east and wind speed above the daily average easterly wind of 10.6 $m\ s^{-1}$, the measured $\delta^{18}O$ values are higher than the predicted $\delta^{18}O$ value. This indicates that even for days, when no strong temperature changes can be observed, strong winds from east coincide with more enriched $\delta^{18}O$ values in water vapour at Neumayer Station. On the opposite, for 76 % of the days with katabatic winds and a wind speed higher than the daily averaged southerly wind of 4.6 $m\ s^{-1}$ (12 out of 17 days), measured $\delta^{18}O$ values are lower than the predicted $\delta^{18}O$ values. This means that southerly winds can transport water vapour with a more depleted $\delta^{18}O$ composition to the station, and such transport might occur without any significant temperature change.

## 4.2 Key controls on vapour $d$

Changes of $d$ in vapour generally are supposed to reflect different climate conditions at the moisture source region (Merlivat and Jouzel, 1979; Pfahl and Sodemann, 2014) and recognized as a tracer that point out the origin of the water vapour (Gat,

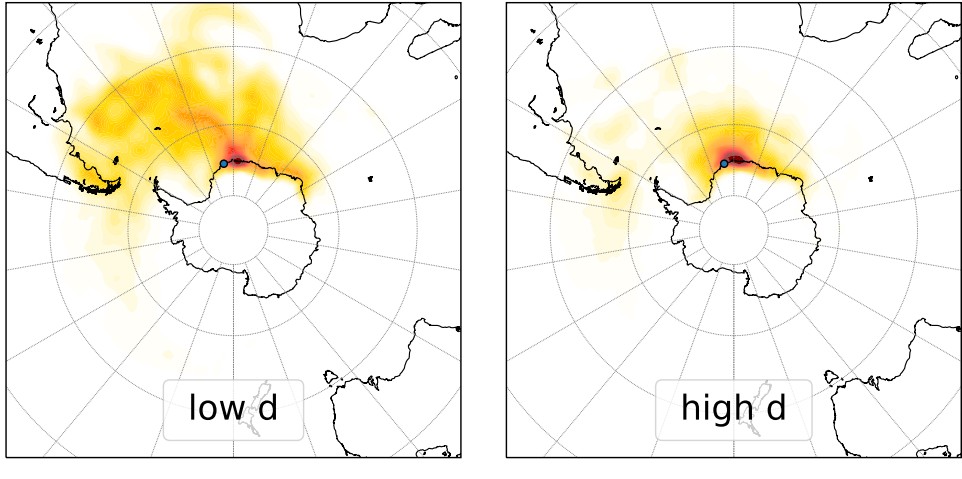

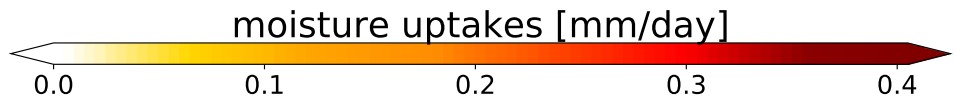

**Figure 11.** Simulated moisture uptake occurring within the boundary layer [ $mm\ day^{-1}$] towards Neumayer Station modelled by FLEX-PART for the whole period of the experiment, left: an average on days with low $d$ events, right: an average on days with high $d$ events.

1996). Other, less dominant factors that effect $d$ are the amount of condensation from source to sink, which is ruled out here since there is no systematic anti-correlation between $d$ and $\delta D$.

In order to better understand the effect of different pathways and water moisture origins that control $d$ changes in vapour at the station, days with extreme values of $d$ are examined. As no long-term measurements for $d$ in water vapour at Neumayer

Station exist, we cannot define extreme $d$ values considering multi-year daily average (as done for the analysis of extreme temperature events). Thus, we define extreme $d$ values as daily averaged $d$ values which are one standard deviation higher or lower than the 14-days average value centered around the corresponding day (7 days before and 7 days after). Analysing the simulated moisture uptake for days with low and high $d$ values reveals the influence of the origin of the water vapour on extreme $d$ values (Fig. 11. The moisture corresponding to low daily $d$ values is either uptaken in coastal areas east of the

station (this occurs mostly in summer) or north-west of it in the South Atlantic Ocean. For such low $d$ values of a marine moisture origin, sea surface temperature and near-surface relative humidity are the prime controls (Merlivat and Jouzel, 1979). The occurrence of high $d$ in water vapour originating from a sea surface area close to a sea ice margin has been explained by different studies such as Kurita (2011) and Steen-Larsen et al. (2013). Based on the back trajectory simulations, the high $d$ values in our measurements can be explained in a similar way. Strong evaporation from sea surface waters into cold humidity-

depleted polar air close to the ice-margin (here, areas close to Neumayer Station) will result in strong kinetic effects providing higher $d$ values in the water vapour.

## 4.3    Comparison to other sites

To further assess the results of our water stable isotopes measurements in vapour at Neumayer Station, we look at comparable vapour measurements in Antarctica. To our knowledge, our study is the first one measuring water isotopes in vapour in
Antarctica throughout the whole year. Other studies have been performed only during austral summer (mostly December and January), for limited periods of 40 days or less (Ritter et al., 2016; Casado et al., 2016; Bréant et al., 2019). Thus, for the comparison of our data with these previous studies, we focus on our austral summers results (December 2017/January 2018 and December 21018/January 2019), only.

### 4.3.1    Dumont d'Urville station in Adélie Land

Most comparable to our measurements is a recent study by Bréant et al. (2019), who have reported water isotopes in vapour from the Dumont d'Urville station in Adélie Land, which is also located at the Antarctic coast (Fig. 1). The station is located on Petrel Island which is about 5 km from the continent and the ice sheet. During summer, most of the island is free of ice and snow and sea ice is rare (König-Langlo et al., 1998). The average temperature at the Dumont d'Urville station during the study period was -0.5°C and the average value of $\delta^{18}O$ in vapour was -30.37 ‰. At Neumayer Station, $\delta^{18}O$ values in
vapour related to a comparable summer temperature (-0.5 $\pm 0.05$°C) are substantially more enriched ($-24.47 \pm 2.34$ ‰). The lower $\delta^{18}O$ values at the Dumont d'Urville station can be explained by the prevailing katabatic winds at this location, which bring dry, depleted air from the continent to the Dumont d'Urville station. For the relationship between $\delta^{18}O$ and temperature, the correlation coefficient for Dumont d'Urville is $r = 0.49$ for clear-sky conditions and $r = 0.33$ for cloudy-sky conditions. Our all-sky summer value at Neumayer Station is $r = 0.55$. For better comparison with Dumont d'Urville station, we also
calculated different values for cloudy and clear-sky conditions at Neumayer Station. To determine days with cloudy or clear sky at Neumayer Station, we used the same method as in Bréant et al. (2019). To do this, all days with the median long wave downward radiation values higher (lower) than 280 $W.m^{-2}$ are considered as days with a cloudy-sky (clear-sky). Our study shows a slightly higher correlation coefficient than at Dumont d'Urville station for both conditions at Neumayer Station (clear-sky: $r = 0.60$, cloudy-sky: $r = 0.53$). Bréant et al. (2019) argue that temperature cannot be a main driver for $\delta^{18}O$ in water
vapour due to this low correlation coefficient, even for the clear-sky conditions. However, the correlation coefficient between $\delta^{18}O$ in vapour and specific humidity is high ($r = 0.85$) at Dumont d'Urville, but only for clear-sky conditions. When the sky is cloudy, quantities seem to be totally uncorrelated ($r < 0.01$). For Neumayer Station, this correlation exists for both conditions similarly, but lower than the one at Dumont d'Urville in clear-sky conditions (at Neumayer Station; for clear-sky: $r = 0.59$, for cloudy-sky: $r = 0.60$). At the Dumont d'Urville station, the correlation between $d$ and specific humidity in vapour is $r = -0.71$
for clear-sky conditions, while they are uncorrelated ($r < 0.01$) for cloudy-sky conditions. At Neumayer Station, $d$ and specific humidity in vapour are anti-correlated under cloudy-sky conditions ($r = -0.55$), but they are weakly correlated for clear-sky conditions ($r = -0.31$).

To analyse changes during the diurnal cycle, Bréant et al. (2019) categorized their measurements based on the weather conditions as days with a clear sky and days with a cloudy sky. Since we do not find large differences in diurnal cycles related to these two conditions at Neumayer Station, we do not cluster our measurements. At both stations, strong diurnal cycles in $\delta^{18}O$, $\delta D$, $d$, temperature and specific humidity are observed. The diurnal cycle of the mostly katabatic winds is stronger at Dumont d'Urville station than Neumayer Station. At Dumont d'Urville station, a relatively low correlation between the diurnal cycle of temperature and $\delta^{18}O$ shows that the temperature cannot be the main driver of $\delta^{18}O$ variations, while at Neumayer Station it is the main driver of diurnal variations of $\delta^{18}O$. At Dumont d'Urville station, diurnal cycle links start with the temperature variations in the continental areas above the station due to the incoming shortwave radiation diurnal cycle. A decrease of incoming shortwave radiation leads to radiative cooling at the continental slopes surface (above the station), resulting in an increase of the katabatic wind, which is characterized by lower $\delta^{18}O$ and higher $d$ values and thus causes the diurnal cycles at Dumont d'Urville station. At Neumayer Station, the shortwave radiation affects mainly the local temperature results in $\delta^{18}O$ value variations, while the dominant wind, in the absence of the strong katabatic wind, is from east.

At both stations, cold and dry air parcels from the continent, which are transported by katabatic winds to the stations, are characterized by low specific humidity, low $\delta^{18}O$, and high $d$ values. At Dumont d'Urville station, katabatic winds are the prevailing wind system due to its complex topography. They are on average considerably stronger than the katabatic winds observed at Neumayer Station and often lead to clear-sky conditions. In these situations, the amount of moisture transported by the katabatic winds to Dumont d'Urville station determines the $\delta^{18}O$ values in vapour. However, for Neumayer Station, the prevailing wind system is easterly winds bringing humid air with relatively high $\delta^{18}O$ values from the Southern Ocean to the station. But for days with katabatic winds at Neumayer Station, the effect of these cold, dry winds on the water stable isotopes composition in vapour appears to be qualitatively comparable to the situation at Dumont d'Urville station.

### 4.3.2  Kohnen Station, Dronning Maud Land

The location closest to Neumayer Station, at which vapour isotope measurements have been performed, yet, is the Kohnen station in Dronning Maud Land, with a distance of 757 km inland, SSE to Neumayer Station (Fig. 1). Kohnen Station is located in a continental area of Antarctica at an elevation of 2892 m (Ritter et al., 2016). Water vapour isotopes were measured at this station for one month during the austral summer season 2013–2014 (Ritter et al., 2016). The mean temperature during the study period at Kohnen Station was -23.40°C which is approx. 20°C lower than the temperature at Neumayer Station during the same period. In accordance with this lower mean temperature, the water vapour at Kohnen Station was characterized by a lower mean $\delta^{18}O$ signal (-54.74 ‰) than the one at Neumayer Station ($-39.11 \pm 4.08$ ‰ for considered temperature, -23.40°C). At Kohnen Station $\delta^{18}O$ is correlated with specific humidity ($r = 0.56$) and with temperature ($r = 0.49$). These correlation coefficients at Neumayer Station (for temperature–$\delta^{18}O$, $r = 0.63$, and for specific humidity–$\delta^{18}O$, $r = 0.68$), are higher than the respective values found at Kohnen Station during the 1 month measuring period, though. At both Neumayer Station and Kohnen Station, $\delta^{18}O$ and $\delta D$ are also highly correlated ($r = 0.98$ and $r = 0.97$, respectively), but the slope between these variables at Neumayer Station is larger than the slope at Kohnen Station. Neumayer Station is a coastal station while Kohnen Station is a continental station. Although the two stations are relatively close to each other, they have very

**Table 1.** Comparison of summer water vapour isotope studies in different Antarctic stations (Ritter et al., 2016; Bréant et al., 2019). The time period of each study is shown in the table. For Neumayer Station, two months of two summers (December 2017/January 2018 and December 21018/January 2019) are considered.

| | Neumayer | Dumont d'Urville | Kohnen |
|---|---|---|---|
| **location** | coastal | coastal | inland |
| **period of measurements** | 01/12/2017 - 31/01/2018 and 01/12/2018 - 19/01/2019 | 26/12/2016 - 03/02/2017 | 17/12/2013 - 21/01/2014 |
| $\delta^{18}O$ **vs. temperature** | $r_{clear} = 0.60 \quad r_{cloudy} = 0.53$ $r_{total} = 0.63$ | $r_{clear} = 0.49 \quad r_{cloudy} = 0.33$ | $r = 0.49$ |
| $\delta^{18}O$ **vs. specific humidity** | $r_{clear} = 0.58 \quad r_{cloudy} = 0.60$ $r_{total} = 0.68$ | $r_{clear} = 0.85 \quad r_{cloudy} \approx 0$ | $r = 0.56$ |
| $d$ **vs. specific humidity** | $r_{clear}=-0.31 \quad r_{cloudy}=-0.55$ $r_{total} = -0.55$ | $r_{clear} = -0.71 \quad r_{cloudy} \approx 0$ | -no data- |
| $\delta^{18}O$ **vs.** $\delta D$ | $r = 0.98, s = 6.7\,‰\,‰^{-1}$ | -no data- | $r = 0.97 \; s = 6.2\,‰\,‰^{-1}$ |
| **main weather pattern** | easterly wind | katabatic wind | katabatic wind |
| **Diutnal cycle parameter** | Neumayer | Dumont d'Urville | Kohnen |
| **diurnal cycle period** | all days | only clear sky periods | 20/12/2013 - 27/12/2013 |
| **temperature [°C]** | average value: -4.24 diurnal cycle amplitude: 4.14 | average value: -0.5 diurnal cycle amplitude: 4.5 | average value: -23.4 diurnal cycle amplitude: 8.7 |
| **specific humidity** $[g\,kg^{-1}]$ | average value: 2.49 diurnal cycle amplitude: 0.54 | average value: 1.99 diurnal cycle amplitude: 0.80 | average value: 0.74 diurnal cycle amplitude: 0.62 |
| **wind** $[m\,s^{-1}]$ | average value: 7.5 diurnal cycle amplitude: 1.5 | average value: 8 diurnal cycle amplitude: 7 | average value: 4.5 diurnal cycle amplitude: 3.5 |
| $\delta^{18}O$ **[‰]** | average value: -26.34 diurnal cycle amplitude: 2.45 | average value: -30.37 diurnal cycle amplitude: 5.40 | average value: -54.74 diurnal cycle amplitude: 4.87 |
| $d$ **[‰]** | average value: 5 diurnal cycle amplitude: 5 | average value: 3 diurnal cycle amplitude: 10 | average value: 30 diurnal cycle amplitude: 11 |

different climate conditions. At Kohnen Station, strong katabatic winds dominate the changes in isotopic values of water vapour, while at Neumayer Station winds from east related to cyclonic weather pattern centered over the Southern Ocean play a major role (Medley et al., 2018).

Ritter et al. (2016) chose 18 days of their available measurements to evaluate the diurnal cycles at Kohnen Station. They observed a strong diurnal cycle in temperature, specific humidity, wind speed (mostly katabatic), $\delta^{18}O$, $\delta D$ and $d$, while at Neumayer Station, a weak daily cycle of wind changes (mostly from east) is detected. For the diurnal variations at Kohnen Station, there exist high correlations between specific humidity and $\delta D$ ($r = 0.99$) and also between temperature and $\delta D$ ($r = 0.99$) which highlight the role of temperature in $\delta D$ and $\delta^{18}O$ diurnal cycles. In contrast to Neumayer Station, the $d$ values

are strongly anti-correlated with $\delta^{18}O$ and $\delta D$ at Kohnen Station. Ritter et al. (2016) suggested that this strong anti-correlation is caused by a distillation effect at very low temperatures (mean temperature at Kohnen station during their campaign is -23.40 °C).

    The information of the available summer data sets from these three locations (Neumayer Station, Dumont d'Urville, Kohnen Station) are summarized in Table 1.

**4.4   Comparison of water vapour and snow isotope ratios at Neumayer Station**

We compare our vapour measurements to isotope measurements of snow in the vicinity of Neumayer Station. At Neumayer Station, fresh snow has been sampled after major snowfall events since 1981. Such snowfall events usually coincide with strong winds and snow samples are normally taken from fresh precipitation combined with blowing snow (Schlosser, 1999). In contrast to locations in the interior of Antarctica, snowfall events in coastal areas of the Ekström Ice Shelf, where the station

is located, are evenly distributed over the whole year (Helsen et al., 2005). Mean isotope values of these snow samples from Neumayer Station are -20.54 ‰ for $\delta^{18}O$ and 153.25 ‰ for $\delta D$, for the period 1981 to 2000 (Schlosser et al., 2004). The slope between the 2-m temperature and $\delta^{18}O$ in the snow samples was determined as 0.57 ‰ $°C^{-1}$ ($r = 0.69$) for the period 1981 to 2000 (Schlosser et al., 2004). This value was confirmed by Fernandoy et al. (2010), who found an identical T–$\delta^{18}O$ slope for snow samples of the extended period from 1981 to 2006. This reported slope in snow samples is very close to our findings for

the relation between $\delta^{18}O$ in water vapour and the 2-m temperature (0.58 ‰ $°C^{-1}$, $r = 0.89$). Additionally, we find a similar slope between $\delta^{18}O$ and $\delta D$ in our vapour measurements (7.67 ‰ ‰$^{-1}$) for the period February 2017 to January 2019, as it has been reported for snow samples (7.95 ‰ ‰$^{-1}$) for the period 1981 to 2006 (Fernandoy et al., 2010). Schlosser et al. (2004) found that the T–$\delta^{18}O$ slope for the snow samples at Neumayer Station varies between 0.43 ‰ $°C^{-1}$ and 0.75 ‰ $°C^{-1}$, depending on the origin of the moisture and the transport path. By performing a back-trajectory analysis, the authors showed

that a lower T–$\delta^{18}O$ slope in the snow samples corresponds to air parcels with an origin over the ocean, while higher T–$\delta^{18}O$ slopes are related to air parcels originating from continental areas. We consider in our vapour isotope measurements the data for only two years (February 2017 to January 2019) and calculate T–$\delta^{18}O$ slopes for different seasons of the year. The slopes range from 0.48 ‰ $°C^{-1}$ to 0.68 ‰ $°C^{-1}$, with the lowest slope occurring in austral winter for air parcels with a oceanic origin in lower latitudes. Higher slopes are found in austral summer, and our FLEXPART analyses reveal that the summerly

water vapour originates in nearby coastal areas (the oceanic and continental areas close to the station).

Recently, several studies have reported an exchange of water isotopes between surface snow and the vapour above the surface for Greenland (Steen-Larsen et al., 2014a; Madsen et al., 2019) and Antarctica (Casado et al., 2018). Steen-Larsen et al. (2014a) showed in their case study in Greenland, that surface snow $\delta^{18}O$ between snowfalls events, follows $\delta^{18}O$ in water vapour with similar or smaller changes than the water vapour $\delta^{18}O$. The authors suspect that surface snow isotopes are driven by changes in the water vapour isotopic compositions. In our study, the agreement in the T–$\delta^{18}O$ slopes and also in the $\delta^{18}O$–$\delta D$ slopes for water vapour and snow samples measured at Neumayer Station is remarkable. Thus, the isotopic exchange between water vapour and surface snow at this location should be further examined in more detail in future research studies. Since our study provided water vapour isotopic composition measurements with a high temporal resolution and the water vapour isotope monitoring is continuously running at Neumayer Station, we plan to compare the isotope data of water vapour and snow in more detail in the future. If surface snow isotopic compositions are derived by changes in water vapour isotopes, ice core water isotope records might be interpreted as continuously recorded paleoclimate signals, even for periods without any precipitation.

## 5 Conclusions

In this study we analyse the first continuous measurements of water vapour isotopic composition at Neumayer Station, Antarctica, over a period of two entire years (February 2017 to January 2019). This unique data set makes it feasible to study not only summer conditions, but seasonal differences and variations of water vapour isotopes at an Antarctic station. Our measurements reveal a clear seasonal cycle of $\delta^{18}O$ and $\delta D$ variations in vapour, with maxima at the end of austral summer and minima in austral winter. The variability of $\delta^{18}O$ and $\delta D$ is hereby strongly driven by changes in local temperature and specific humidity, both on seasonal and shorter time scales. Generally, a stronger isotopic depletion in vapour at Neumayer Station can be associated with colder, dryer air from the south, whereas the vapour is isotopically enriched, when relatively warm easterly winds prevail. However, our data set does not clearly reveal a seasonal cycle for the Deuterium excess in vapour. This may be due to the low accuracy of the measurements in winter, when the specific humidity at Neumayer Station is extremely low. Despite this measurement uncertainty, $d$ shows a slightly stronger correlation with specific humidity than with temperature. Considering two months of two sequent summers (December-January of 2017/18 and 2018/19), strong diurnal cycles in $\delta^{18}O$, $\delta D$, $d$, 2-meter temperature, 10-meter temperature, specific humidity, and relative humidity are detected. As a result of the incoming shortwave radiation cycle, the diurnal cycle of temperature is identified as the main driver of water vapour $\delta^{18}O$ and $\delta D$ diurnal variations.

Similar findings as for Neumayer Station have been reported for Dumont d'Urville and Kohnen Stations, where also specific humidity was better correlated with $\delta^{18}O$ than temperature, however, for completely different reasons than at Neumayer Station. Whereas Dumont d'Urville is strongly influenced by katabatic winds, which advect cold and isotopically depleted air from the continent to the base, Kohnen Station is situated on the East Antarctic Plateau in a dry and cold climate very different from conditions at Neumayer Station. While temperature changes play a main role in diurnal cycles at Neumayer Station and Kohnen Station, in Dumont d'Urville, katabatic wind is the key control for driving diurnal cycles.

A comparison of vapour and snow isotope data shows that the slopes of both the temperature–$\delta^{18}O$ relationship and the $\delta^{18}O$–$\delta D$ relationship is remarkably similar for snow and vapour at Neumayer Station. This similarity draws more attention to recent findings, which show changes of water vapour isotopes may be a key driver of variations in surface snow isotopes. Such control could explain how paleoclimate signals can be continuously recorded by water isotope variations in ice cores, even for periods without any precipitation. It has been suggested that a local diurnal cycle of sublimation and deposition could cause this isotope exchange between vapour and snow, but more detailed case studies and a combination of vapour and snow isotope data are required to better understand this process.

The performed moisture source diagnostics based on back-trajectory simulations show that the moisture origin for Neumayer Station depends on the seasons regarding the sea ice extent, semi-annual oscillation, and absolute temperature. With the frontal zone moving north in summer, moisture uptake for Neumayer is closer to the coast in summer, whereas in spring and fall the moisture has its origin in a wider region reaching farther north. However, the presented back-trajectory simulations are least reliable for cold periods due to a strong bias in the ERA5 data set concerning wind speed and the frequency of southerly winds. This particularly affects our analysis based on back-trajectory simulations when cold, dry and thus, depleted air is advected from the continent. Also, the very high variability of surface pressure around Antarctica has to be considered, and a longer study period would be desirable in order to get more reliable results.

The Picarro measurements at Neumayer Station are currently being continued and supplemented by surface snow sampling. They will be used for a longer-term study in the future, which should also help to confirm and support the results of this study in more detail.

## Appendix A: Calibration program for water vapour isotope measurements

To apply the calibration, we defined a calibration protocol modifying the protocol developed by Steen-Larsen et al. (2013) and Bonne et al. (2014). Based on the protocol we need to correct the isotopic observation in four aspects: the humidity concentration dependence of isotopic measurements, the potential long-term drift of the instrument, the offset between measured and real isotope values, and the wrong measurements related to special events.

To calibrate the instrument, we need to measure different known water stable isotopes in different conditions. One of the modifications of the initial protocol is related to the method of producing water vapour for the calibration. We consider 4 glass bottles (named "bubbler" with a number from 1 to 4) of water standards with known isotopic compositions, kept at a constant temperature. The water vapour with adjustable humidity is made by blowing dry air into the bubblers and making bubbles. For Picarro L2140-i the range of water concentration is originally defined between 1000 and 50000 ppm. For humidity values below 2000 ppm, systematic instrumental errors in the measured isotope values need to be corrected by so-called humidity response functions (see Steen-Larsen et al., 2014b, for details). In order to determine the humidity response functions of our calibration setup, we measure different known isotopic compositions over a range of water concentrations between 100 and 10000 ppm and computed the humidity response functions as the interpolation of the distribution of all measurements for each standard, using a polynomial function of $2^{nd}$ order (Fig. A1). Humidity response sequences are measured once per year

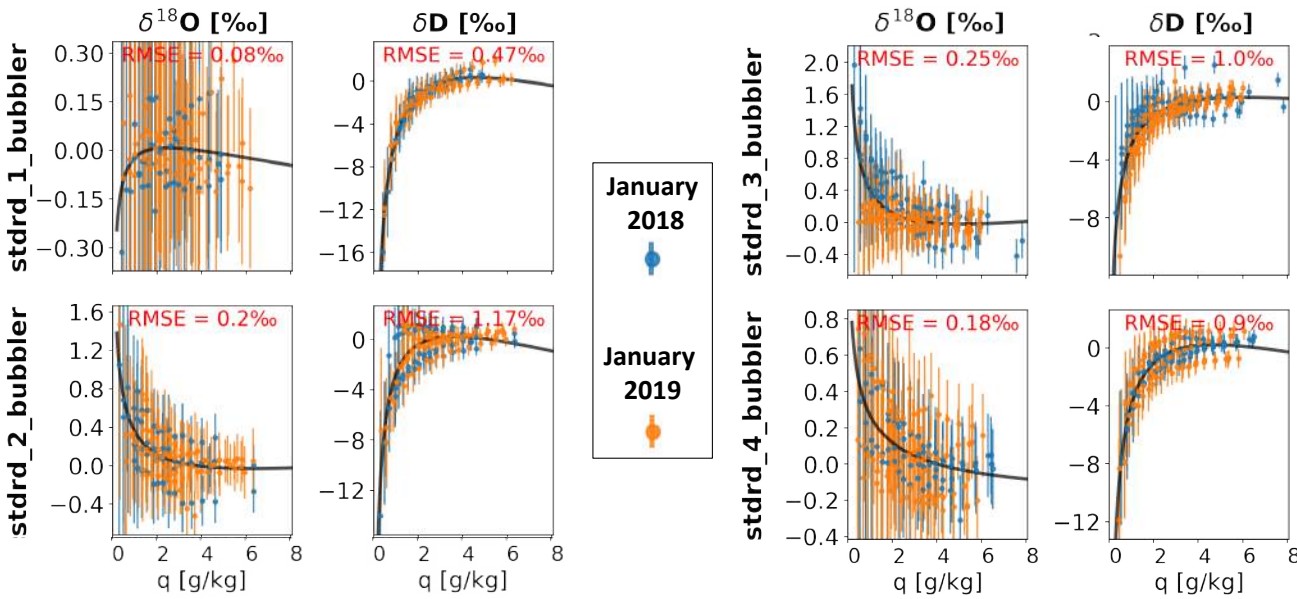

**Figure A1.** Humidity response function curves for different standards at different specific humidity levels for $\delta^{18}O$ and $\delta D$. The isotope measurements and their uncertainties (light blue lines) are plotted as anomaly values, calculated in relation to the mean value of the isotopic composition for every bubbler. The values are fitted by a second degree polynomial curve (dark blue line).

and for this study the humidity response functions of two years (January 2018 and January 2019) have been considered. The calibration system at Neumayer Station was also modified based on the isotopic composition of water vapour at Neumayer Station and used 3 different isotopic water standards (liquid), with $\delta^{18}O$ values of $-6.07 \pm 0.1$ ‰ (around $-17$ ‰ in water vapour), $-25.33 \pm 0.1$ ‰ (around $-36$ ‰ in water vapour), and $-43.80 \pm 0.1$ ‰ (around $-54$ ‰ in water vapour). $\delta D$ values of the standards (water liquid) are $-43.73 \pm 1.5$ ‰, $-195.21 \pm 1.5$ ‰, and $-344.57 \pm 1.5$ ‰. One of the isotope standards ($\delta^{18}O = -25.33$ ‰) is used for quality control in not one but two of the four bubblers. Every year in January, a sample of each standard is taken and transferred to a laboratory in AWI Bremerhaven and was measured in order to know the isotopic composition. In this study, no change in the isotopic compositions of the standards has been detected. Water standards with water vapour $\delta^{18}O$ values of around $-54$ ‰, $-36$ ‰, and $-17$ ‰, cover the whole isotopic measurements range in water vapour at Neumayer Station. The correction of humidity dependant water vapour isotopic measurements has been done by a linear interpolation of those two moisture response functions that belong to the isotopic standards above and below each measured isotopic value. After applying the humidity response calibration, the correction of the measurements based on the instrumental drift and offset is done. We measure the isotopic composition of all water standards every 25 hours. Such 25-hour cycle avoids repetition of the daily calibration at the same time every day but moves the calibration time 1 hour forward per

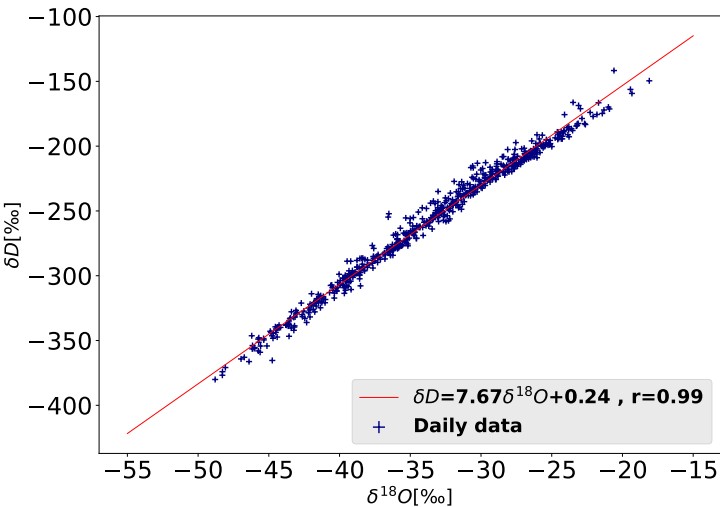

**Figure B1.** Daily averaged observed $\delta^{18}O$ [‰] vs. $\delta D$ [‰] at Neumayer Station from February 2017 to January 2019. A best fitted line, using the least-squares approach, is plotted as a red line and corresponding correlation coefficients are calculated.

day. By such a calibration interval, we avoid missing the same time period during all days in all measurements. Each isotopic standard is measured for 30 minutes, and to avoid any memory effects, we consider the mean value of the last 15 minutes of each measured water standard, only. If the calibration measurements are not stable enough, we filter them. To correct the data based on the instrumental drift, we consider the average of 14 days of measurements, 7 days before and 7 days after the day that we want to correct its data. The correction of the instrumental drift and offset are done by a linear interpolation of the two closest mean isotopic standards, measured at the 25-hour interval.

**Appendix B: Extra tables and figures**

**Table B1.** The monthly mean values and uncertainty (calculated from hourly data) for $\delta^{18}O$, $\delta D$, and $d$ from February 2017 to January 2019.

| | $\delta^{18}O$ ‰ | | $\delta D$ ‰ | | **d** ‰ | |
|---|---|---|---|---|---|---|
| **Month** | mean | uncertainty | mean | uncertainty | mean | uncertainty |
| 2017-2 | -27.98 | 0.32 | -205.33 | 2.45 | 18.52 | 2.48 |
| 2017-3 | -33.30 | 0.50 | -244.85 | 3.57 | 21.54 | 3.61 |
| 2017-4 | -31.66 | 0.33 | -231.04 | 2.35 | 22.21 | 2.38 |
| 2017-5 | -36.78 | 0.51 | -277.82 | 3.33 | 16.43 | 3.40 |
| 2017-6 | -39.03 | 0.66 | -298.03 | 4.08 | 14.18 | 4.14 |
| 2017-7 | -40.72 | 0.69 | -311.68 | 4.43 | 14.08 | 4.49 |
| 2017-8 | -39.56 | 0.51 | -306.20 | 3.29 | 10.27 | 3.34 |
| 2017-9 | -35.65 | 0.47 | -273.79 | 3.13 | 11.40 | 3.17 |
| 2017-10 | -32.10 | 0.27 | -247.20 | 1.90 | 9.57 | 1.92 |
| 2017-11 | -29.64 | 0.28 | -227.72 | 1.86 | 9.40 | 1.88 |
| 2017-12 | -26.88 | 0.33 | -209.62 | 1.88 | 5.38 | 1.90 |
| 2018-1 | -26.35 | 0.34 | -207.00 | 1.98 | 3.77 | 2.00 |
| 2018-2 | -29.94 | 0.57 | -228.65 | 2.46 | 10.91 | 2.49 |
| 2018-3 | -25.83 | 0.42 | -200.33 | 2.48 | 6.30 | 2.51 |
| 2018-4 | -33.04 | 0.58 | -256.36 | 3.77 | 7.96 | 3.82 |
| 2018-5 | -30.90 | 0.42 | -239.45 | 2.71 | 7.77 | 2.74 |
| 2018-6 | -36.94 | 0.58 | -285.62 | 3.88 | 9.91 | 3.92 |
| 2018-7 | -40.21 | 0.48 | -312.19 | 2.87 | 9.50 | 2.91 |
| 2018-8 | -41.14 | 0.64 | -319.79 | 3.75 | 9.29 | 3.81 |
| 2018-9 | -37.74 | 0.49 | -290.86 | 3.08 | 11.09 | 3.12 |
| 2018-10 | -37.73 | 0.56 | -287.14 | 3.53 | 14.66 | 3.58 |
| 2018-11 | -31.50 | 0.37 | -240.29 | 2.50 | 11.73 | 2.53 |
| 2018-12 | -27.34 | 0.52 | -208.38 | 3.46 | 10.34 | 3.50 |
| 2019-1 | -24.70 | 0.41 | -193.01 | 2.47 | 4.56 | 2.50 |

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

*Competing interests.*

We have no conflicts of interest to disclose.

*Acknowledgements.* This work was funded by the Helmholtz Climate Initiative REKLIM (Regional Climate Change), a joint research initiative of the Helmholtz Association of German research centres (HGF).