# Peer review of "Continuous monitoring of surface water vapour isotopic compositions at Neumayer Station III, East Antarctica"

_The Cryosphere, 2020_

## Referee Comment (RC1) · Anonymous Referee #1 · 17 Nov 2020

This manuscript presents the first continuous monitoring of surface vapour isotopic composition at Neumayer over two years. It is the first time that such a record covering two whole years is obtained in Antarctica which makes this study of interest for understanding the isotopic composition of water in Antarctica. The author completed this analysis by an analysis of the moisture source for the Neumayer III over the year 2017 using the Flexpart back-trajectory approach.

Despite the interest of such study and of the new and original dataset, this manuscript still needs to be strongly improved to be acceptable for publication.

Major comments:

[Figure]

- The data are presented but the calibration system is not well described and it is questionable to know if it is adapted to the Neumayer conditions. It would be nice to display the calibration curves obtained to correct for the instrumental drift (down to 100 ppm according to the text) as well as the influence of humidity on water isotopic ratio in the humidity range studied here. A section dedicated to calibration should be added in the supplementary material.

- It is not clear why the diurnal cycles are not shown in this study while they are expected to bring interesting information, especially when comparing to other sites as performed in sections 4.4.1 and 4.4.2. It is thus important that the diurnal cycles (at least in summer) are shown and thoroughly discussed. Otherwise, the comparison performed on section 4.4 on the correlation of the isotopic signal does not make sense since they are done on different timescales.

- The slope of the relationship between q(picarro) and q(meteorology) is very high (1.5). This is really surprising. It would be better to have a direct calibration of the humidity of the picarro (in laboratory using a dew point generator for example).

- Some d-excess peaks look strange and are not discussed (e.g. negative peak in August or September 2018). Also the authors should explain why some periods are lacking in the record.

- I do not see the interest of looking at the d18O vs temperature, then d18O vs ln(q) and then temperature vs ln(q). . . .. The link between temperature and ln(q) can be mentioned in one sentence early in the manuscript stating that the Clausius-Clapeyron relationship is dominating the humidity signal at Neumayer and then remove Figure 5 and Figure 9.

- The appendix with the d-excess correlations is not useful – you could simply detail what is statistically robust or not in the correlation in section 3.2.3.

- I do not understand the sentence on l. 285 -> do you have any measurements (isotopes, temperature, humidity . . .) over some characteristic events showing that d18O

values drop intensively in a distance of 16 km between open sea and the station ? If not, such affirmation should be removed.

- I am not sure that the Figure 11 corresponds very well to the text. The association between wind from the East and higher d18O only works for 22 out of 36 days (61% - not sure to understand where does the 64% value comes from) which is not a proportion strong enough to drive such conclusion on Figure 11. The signal is increased when considering only extremely warm and cold days

- It is not clear which types of events the authors want to discuss in section 4.2 for the key controls on vapour d-excess. Indeed, synoptic events lasting several days and associated with a particular atmospheric pattern will not necessarily be detected by the proposed algorithm: if several following days have a particularly high d-excess values then the average value over 10 days may also be high and the period will not be identified. Moreover, the d-excess pattern found with this definition are not very clear and no clear explanation of the pattern is provided (observed correspondence apply to 70% or less of observed cases). This section is thus not very strong.

- The section 4.3 should be placed earlier (not in the discussion section)

- Section 4.4 cannot stand like this without showing also daily variability at Neumayer

- The section 4.5 aims at discussing the air-snow interaction but the fresh snow is sampled only after major snowfall events which is very different to the other study discussed here (Steen-Larsen et al., 2014) in which snow was sampled at high resolution in presence or absence of precipitation. It is thus impossible to discuss the link between air and snow in the case presented here when surface snow is only sampled after large snowfall events. In this case surface snow isotopic composition reflects the isotopic composition of the precipitation.

Summarizing, the whole discussion section is rather weak and additionnal work should be done on it to make a better use of the new data set.

---

## Referee Comment (RC2) · Anonymous Referee #2 · 21 Dec 2020

Review of Saeid Bagheri Dastgerdi et al.: Continuous monitoring of surface water vapour isotopic compositions at Neumayer Station III, East Antarctica

Bagheri Dastgerdi and co-authors present a new dataset of vapor isotopic composition, sampled continuously at Neumayer Station, Antarctia over two years. The authors analyze the dataset using meteorological observations and back-trajectory modeling. The dataset appears to be of high quality, and sheds light on isotopic processes in the continent important for the interpretation of ice cores. Unfortunately the analysis is somewhat limited. For example, the authors only perform back-trajectory modeling for less than half their dataset, which limits their ability to interpret the data. Some

of the more interesting features – such as a secular trend in the d-excess data – are not addressed. The reader is left with very little new insight in what type of dynamical processes may be driving the isotopic trends.

I am not a meteorologist, but it appears there are some shortcomings in the interpretation of the various weather data (as noted in the detailed comments below). The paper would benefit from being read/reviewed by an expert on Antarctic meteorology. Very few, if any, papers on this topic are cited. Various meteorological observations must be placed in a broader context of such observations, and the correct interpretations must be given.

While somewhat unsatisfying, I believe the paper deserves to be published based on the quality of the unique new observations.

Comments:

The back-trajectory provides the most meaningful and relevant data analysis tool in the paper, but it is only used vey summarily. First, I would encourage the authors to run the backtrajectory on the full data period. Second, I would encourage the authors to provide more in-depth analysis of what the back-trajectories mean.

For example, they show in Fig. 12 that there is a difference in vapor origin between low-d and high-d events. However, the authors do not provide us any insight into WHY these patterns may lead to the observed d values.

For the d-excess, it would be very valuable to understand the long term trend. Is it related to changes in the vapor origins? And can these long-term changes be understood in terms of the large-scale atmospheric circulation, for example through changes in southern annular mode? Or perhaps it is linked to local effects such as the sea ice extent in the Atlantic sector of the Southern Ocean? These types of analysis would provide some valuable insights into the dynamics, which are currently lacking.

Detailed comments:

TCD
Line 7: d-excess; more commonly just d (lower case italicized).

Line 23: Buizert et al. (2015) should be WAIS Divide Project Members (2015)

Line 32: "clouds": much of the precip in central Antarctica is clear-sky precip (diamond dust).

Line 35: temporal relationship \*may\* differ from spatial relationship.

- Line 37: Better citation for inversion strength is (Van Lipzig et al., 2002)
- Line 39: Another key control is sea ice (Noone & Simmonds, 2004)
- Line 72: "sections" instead of "chapters"

Line 92: from the east?

- Line 107: Perhaps define this as the climatology
- Line 109: above (below) the climatology
- Line 113: "merged to"? What about "reported as"

Line 145: what is the philosophy behind the 25h? Is the idea that taking a fixed time of day could introduce biases?

Line 150: Are the tanks measured at the very end of the campaign against independent standards to ensure there was no drift?

Line 162: Why 2017 only? If you had let your code run during the period of the review process, you would probably have these already. Having the full 2017-2019 period would allow you to analyze the d-excess trend. I would strongly encourage you to run these also. It should be little work, given that the code is working.

Line 170: can you add the climatology for comparison (T and q)? Section 2: did the setup require a human operator at all times, or was it fully automated (e.g. to do the calibrations)?
Line 183: Can you also say something about the relative (rather than absolute) humidity? That is a much more intuitive parameter. Perhaps add it to the figure?

Line 192: But the long-term d-excess trend is robust, correct?

Line 196: Not fully reliable given the large difference in slope of 1.5. Is this a calibration issue? Can you elaborate a little – it seems this does not impact your isotope data, but it would be nice to know where this difference in slope originates.

Line 208: Given the strong Antarctic temperature inversion in winter, this 22 m difference may actually matter a lot for temperature and therefore humidity - in the interior of Antarctica, the difference could be up to 10K (Hudson & Brandt, 2005), but the inversion is probably less strong near the coast.

Line 220: How are the seasons defined? Do you use DJF, MAM, JJA, SON?

Line 223: Enhanced temp variability in winter is seen all over Antarctica (e.g. at South Pole), and has been explained elsewhere via the stability of the inversion (Hudson & Brandt, 2005). Please refer to some other papers on the meteorology of Antarctica, I think this is more fundamental and not specific to Neumeyer. I don't think the reason you state is the correct one necessarily. Please clarify and provide relevant citations.

Section 3.2.2.: I think you need to give us the relative humidity plots to evaluate ho meaningful this correlation is. Since q and T are strongly correlated, these observations are unsurprising.

Section 3.2.3.: is there a diurnal cycle in isotopes?

Line 263: Is that really the reason? The cyclonic storm tracks are moving from west to east, so opposite to what you state. I am no meteorologist, but I always thought the near-coastal easterlies were driven by Coriolis deflection of the katabatic winds off the continent. Please clarify and provide relevant citations.

Figure 8: what years of the reanalysis are used?
Figure 8: what is the purpose of this figure? I don't see how it contributes to the paper or the narrative.

Line 283: "the main air path is a cyclonic circulation" What does this mean?

Figure 9: The vapor pressure over ice via the Clausius Clapeyron has an Arrhenius type relationship, with vapor pressure scaling as exp(-H/RT). So not quite the relationship that you show. Can you plot correctly vs. 1/T (with T in Kelvin), and estimate the Enthalpy H?

Line 288: What form of the CC equation do you use? Please give the equation.

L300-302: I don't understand this point. Can we see this in the data?

L305-320: It is not clear what the analysis of the relationship between wind direction and pressure anomaly is based on. Reanalysis data? Case studies? No citations are provided.

L330: Again, this is unfortunate. How long does it take to run the code? Surely not more than a few days?

L345: This is an valuable observation. Does it make sense that water vapor originating close to the continent has higher d-excess? Please elaborate

L352: Is Neumeyer data assimilated in ERA-interim?

L370: So does Neumeyer have greater relative humidity then?

Section 4.5: The agreement in slope between d18O-T from vapor and precip is remarkable. Do you think this relationship is valid only because you are so close to the coast/vapor source?

References:

Hudson, S. R., & Brandt, R. E. (2005). A Look at the Surface-Based Temperature Inversion on the Antarctic Plateau. Journal of Climate, 18(11), 1673-1696. Interactive comment

https://journals.ametsoc.org/view/journals/clim/18/11/jcli3360.1.xml

Noone, D., & Simmonds, I. (2004). Sea ice control of water isotope transport to Antarctica and implications for ice core interpretation. Journal of Geophysical Research: Atmospheres (1984–2012), 109(D7).

Van Lipzig, N. P., Van Meijgaard, E., & Oerlemans, J. (2002). The effect of temporal variations in the surface mass balance and temperature-inversion strength on the interpretation of ice-core signals. Journal of Glaciology, 48(163), 611-621.

---

## Referee Comment (RC3) · Anonymous Referee #3 · 27 Jan 2021

Review for the Cryosphere of

Continuous monitoring of surface water vapour isotopic compositions at Neumayer Station III, East Antarctica

by Saeid Bagheri Dastgerdi and others

General In my opinion, the superb quality of the isotope dataset (with some question marks, see discussion on humidity below) deserves a more in-depth analysis of the atmospheric conditions explaining the isotopic variations in the water vapor. For instance, with reliable ground observations and high resolution (space and time) ERA5 it is now possible to use higher-level temperatures (e.g. 850 hPa) to characterize air

masses in the analysis, see below.

Major Fig. 1 is too empty: use it to indicate summer/winter surface pressure distribution, sea ice edges, surface topography, and grounding line.

Fig. 2: It would be interesting to plot RH separately in this figure as well. How do the moisture measurements of the Picarro instrument compare to those of the meteorological field? Ah, I now see this is reported later (l. 195). Please mention this earlier.

Table 1 can be moved to an appendix.

Fig. 4: In Fig. 3 you show a large difference between q measured at the Picarro inlet and the meteorological field (from T and RH measured at 2 m), and suggest this may be caused by strong near-surface inversions in moisture. Yet, here you continue to directly compare isotope values with temperatures from that same field. Please elaborate somewhat on why you think that does not pose a problem during variable inversion conditions.

l. 195-200: The correlation is high, but a) there is still considerable scatter and b) the slope is off by 50%. These are important differences. These differences can be discussed in somewhat more detail, e.g. are these high values at all possible given the outside temperatures measured at ∼20 m above the surface, or would this imply oversaturation? You could also select non-inversion conditions (cloudy, strong winds) to check your assumptions on the inversions.

Section 3.2.1: Melting and the resulting cutoff of surface temperature at 0 C is probably less relevant to explain the limited day-to-day variability in 2 m temperature during summer, because melting at Neumayer is mostly a daytime feature, i.e. seldomly lasts a full day, and therefore has less impact on daily mean temperature. A more probable explanation is that because of the absorption of solar radiation the surface radiation deficit becomes small or absent in summer, preventing the formation of strong near-surface temperature inversions in the daily mean. It is the regular formation (clear
skies, weak winds) and destruction (cloudy skies, strong winds) of these inversions that explain the large interdiurnal temperature variability in the non-summer seasons.

Fig. 5 and Section 3.2.2: In an environment with unlimited evaporation/sublimation potential such as Neumayer, with near-continuous surface cooling outside of summer, the near-surface air will always be close to saturation (this is what you show in Fig. 9). Repeating the correlation of water isotopes with humidity, as you did in Fig. 4 with temperature, is therefore not so useful.

Section 3.2.3: have you tried correlations with relative humidity?

Fig. 6: please include the sea ice edge. The difference in latitudinal fetch and absolute uptake appears to be a combination of sea ice extent (sea ice preventing evaporation) and the semi-annual oscillation, the twice-annual expansion/contraction of the circumpolar pressure trough which determines the latitudinal fetch. The absolute temperature also plays an important role, with evaporation/sublimation being reduced at low temperatures (winter).

Fig. 8: This figure shows surface pressure reduced to sea level, and is therefore inaccurate over topography. Values over the continent should be masked.

Section 4.1.1: Temperature at Neumayer is mainly controlled by season (determining the free atmosphere, or 'background' temperature) and surface cooling (determining the negative departure of the near-surface temperature from this background temperature). Eliminating the latter by e.g. selecting the 850 hPa temperature over Neumayer from balloon soundings/ERA5 could facilitate the interpretation of water isotope values in terms of airmasses and large-scale circulation.

Minor/textual l.100: "$-16.10$âŮęC ($\pm1.05$âŮę C)". What does the uncertainty indicate here? Given measurement uncertainty, suggest removing one digit, i.e. -16.1 C

l.108: One standard deviation appears rather inclusive. Why was this value selected?

l.115: "For the largest time of the year" -> "For most of the year"

l.124: "Cover the whole range..." but the numbers provided fall outside of that range?

l.138: Relative humidity (RH) is commonly expressed as %, not ppm. Moreover, the latter concentration suggests that you are talking about specific humidity (q). Later on you use 'absolute humidity'. Please clarify and provide these numbers (RH and q) for the presented ppm thresholds as well.

l.182: "Humidity": absolute or specific humidity? l.189: "humidity amount..."Check throughout, please.

l.222: Only the summer slope differs significantly.

l.262: Check the value of the standard deviation, that appears too large.

l. 302: 10.56 -> 10.6, see also l. 305.

l.309: warm/cold temperature -> high/low temperature

l.324: neither -> either

Section 4.4: suggest changing the title into: Comparison to other sites

l.366: 'opposite' please rephrase

Section 4.4.1: Mention that an extensive ice shelf is absent in the case of DDU, and that the station is situated on an island several km away from the coast.

---

## Editor Comment (EC1) · Jean-Louis Tison (Editor) · 28 Jan 2021

Dear Authors,

First of all, please accept my apologies for the relative length of the procedure for this manuscript. As you know (and as I acted previously) the two initial reviewers agreed on "great data set, poor discussion and interpretation". One of the reviewer suggested to request the opinion of a third reviewer with a specific expertise in Antarctic atmospheric processes. This is what I did, and you have seen the comments of this third reviewer recently posted.

As an editor, I was now left with three options: a) request a fourth reviewer's opinion, b) close the discussion and ask for authors final comments and c) reject the paper.

Because of the general opinion that you are providing a unique data set, I have decided for option 2.

I believe all reviewers have gone to very detailed and sound comments that would greatly improve the interpretation of the data, and should be fully taken into account in your final response.

Of course, you still have the option to withdraw your manuscript and submit it to another Journal. If you decide that you have enough arguments to answer all reviewers comments, I will ask you to provide us with:

a) A "one to one" reply to all comments of the 3 reviewers, stating your response and how (and where in the manuscript) you have taken this into account in the new version of the manuscript...(citing the initial version of the paragraph and how you have changed it is of real help for the second round of reviews and the editor for his final assessment)

b) A pristine version of the new version of the manuscript

c) A new version of the manuscript with changes highlighted

On the basis of these documents, and most probably a second round of review(s), I will take my final decision.

Best regards,

Jean-Louis Tison

---

## Author Comment (AC1) · 12 Mar 2021

The comment was uploaded in the form of a supplement:
https://tc.copernicus.org/preprints/tc-2020-302/tc-2020-302-AC1-supplement.pdf

---

## Author Comment (AC2) · 12 Mar 2021

The comment was uploaded in the form of a supplement:
https://tc.copernicus.org/preprints/tc-2020-302/tc-2020-302-AC2-supplement.pdf

---

## Author Comment (AC3) · 12 Mar 2021

**Response to Anonymous Referee #3**

We appreciate the comprehensive, practical, and instructive comments of referee #3. They have helped us improving the paper. We are responding to the comments in the following way:

*General In my opinion, the superb quality of the isotope dataset (with some question marks, see see discussion on humidity below) deserves a more in-depth analysis of the atmospheric conditions explaining the isotopic variations in the water vapor. For instance, with reliable ground observations and high resolution (space and time) ERA5 it is now possible to use higher-level temperatures (e.g. 850 hPa) to characterize air masses in the analysis, see below.*

We did a more in-depth analysis in different aspects such as the examination of the water vapour origin, diurnal cycles in summer, and extreme $d$ values (based on another referee's comment, in the revised manuscript, we use "$d$" instead of "d-excess"). Also, a higher-level temperature (850 hPa) from ERA5 (Hersbach et al., 2020) is evaluated and added to the text (revised manuscript, Subsubsection 3.2.1, line 256). Generally, we have substantially improved the manuscript based on the specific comments of this referee and of two other anonymous referees.

*Fig. 1 is too empty: use it to indicate summer/winter surface pressure distribution, sea ice edges, surface topography, and grounding line.*

We add different information to the figure such as surface topography, mean sea level pressure, Antarctic grounding line, and sea ice edge in austral summer and winter:

Revised manuscript, Subsection 2.1, Figure 1:

[Figure]

*Figure 1. (Figure 1 in the revised manuscript). Map of Antarctica with topography [meter], mean sea level pressure [hPa], Antarctic grounding line (black line), and sea ice fraction [red line: fraction > 0.45, white line: fraction > 0.90] in austral summer (big map) and austral winter (small map), considering years of 2017 and 2018. The topography, mean sea level pressure and sea ice fraction are based on meteorological data from the European Centre for Medium-range Weather Forecasts (ECMWF) ERA5 reanalysis (Hersbach et al., 2020) and the Antarctic grounding line is based on Depoorter et al. (2013). The location of our study (Neumayer III) and other studies which provided continuous water vapour isotopic measurements in Antarctica (Ritter et al., 2016; Casado et al., 2016; Bréant et al., 2019) are shown in white colour and JARE cruise track related to Kurita et al. (2016) is shown in yellow colour.*

*Fig. 2: It would be interesting to plot RH separately in this figure as well. How do the moisture measurements of the Picarro instrument compare to those of the meteorological field? Ah, I now see this is reported later (l. 195). Please mention this earlier.*

The relative humidity and its climatology are added to the figure (Figure 2). The comparison between the humidity measured by Picarro and the one measured by meteorologists at the station is brought up earlier (revised manuscript, Subsection 3.1, line 193).

Revised manuscript, Subsection 3.1, Figure 2:

[Figure]

*Figure 2. (Figure 2 in the revised manuscript). Daily averaged observations at Neumayer Station from February 2017 to January 2019. Downward: a) 2-m temperature [˚C]; b) specific humidity [ g kg⁻¹]; c) relative humidity [%]; d) δ¹⁸O [‰]; e) δD [‰]; f) d [‰]. To have a better comparison, the climatology (multi-year daily average temperature, specific humidity, and relative humidity over the 38-year period from 1981 to 2018) is shown with a red line in meteorological observations. The determined uncertainties of the Picarro instrumental data (see text) are plotted as gray lines.*

*Table 1 can be moved to an appendix.*

Corrected as requested (revised manuscript, Appendix B, Table B1).

*Fig. 4: In Fig. 3 you show a large difference between q measured at the Picarro inlet and the meteorological field (from T and RH measured at 2 m), and suggest this may be caused by strong near-surface inversions in moisture. Yet, here you continue to directly compare isotope values with temperatures from that same field. Please elaborate somewhat on why you think that does not pose a problem during variable inversion conditions.*

*l. 195-200: The correlation is high, but a) there is still considerable scatter and b) the slope is off by 50%. These are important differences. These differences can be discussed in somewhat more detail, e.g. are these high values at all possible given the outside temperatures measured at ~20 m above the surface, or would this imply over saturation? You could also select non-inversion conditions (cloudy, strong winds) to check your assumptions on the inversions.*

We have analysed the different q values now in more detail and expanded the corresponding text in the revised manuscript as follows:

Revised manuscript, Subsection 3.1, line 193:

We compare the specific humidity measured by the Picarro instrument with the specific humidity values measured routinely as part of the meteorological observations at Neumayer Station (Schmithüsen et al., 2019). The relationship between these two series of humidity measurements is $q_{(Picarro)} = 1.5q_{(meteorology)}+0.08$ (N = 12198, hourly values between 17 February 2017 and 22 January 2019, r = 0.97, standard error of the estimate = 0.0022 g.kg$^{-1}$; revised manuscript, Fig. 3). The rather high slope between both humidity measurements and also a number of unusual high and low Picarro humidity values motivated us to analyse the difference between both humidity data sets in more detail.

The inlet of the Picarro instrument is situated approx. 17.5 m above the surface level of the station. As the station is placed on a small artificial hill, this surface level is approx. 7.6 m higher than the surface level of the meteorological mast placed 50 meters besides the station building. Thus, the total height difference between the Picarro inlet and the height of the meteorological humidity measurements is approx. 22 m. In principle, higher humidity values at the Picarro inlet could be explained by a humidity inversion layer above the surface, which could remove near-surface moisture at the meteorological mast position by hoar frost formation. However, temperature differences between a 2-meter temperature sensor at the meteorological mast and temperatures measured on the roof of the station do not exceed 2 °C during our measurement period. No strong temperature inversions are found for the days with extreme Picarro humidity measurements.

To test if contamination by exhaust gases could be another reason for the data mismatch, the wind direction was analysed for those hourly Picarro humidity values which are much higher than the corresponding humidity values measured by the meteorological station. Most of the outliers coincide with a wind direction from the south (and a few from east), which excludes the possibility that a contamination by exhaust gases is the reason for the unusually high Picarro humidity values.

Picarro humidity measurements have been compared with independent humidity observations for a few studies, so far. Aemisegger et al. (2012) calibrated and controlled the humidity of their Picarro instrument by a dew point generator and showed a linear relationship between

Picarro measurements and the humidity measured by the calibration system with a slope of 1.27 and an uncertainty of 100-400 ppm (0.06-0.24 g.kg$^{-1}$). Tremoy et al. (2011) reported that the slope between humidity measured by a meteorological sensor and humidity measured by a Picarro instrument can change from 0.81 to 1.47 depending on site conditions. Bonne et al. (2014) also showed a non-linear response of their Picarro instrument compared with the humidity measured by a meteorological sensor. Based on their data, the ratio between Picarro humidity and sensor humidity values changed from 1 to 1.87, depending on the amount of humidity. Compared to the results of these studies, we rate the calculated ratio of our Picarro humidity measurements versus the humidity data from the meteorological mast ($q_{(Picarro)}/q_{(meteorology)} = 1.5$) as unobstructive.

As in previous studies (e.g., Bonne et al., 2014) we will use the Picarro humidity data for the calculation of the humidity response functions required for the calibration of the isotope measurements. All analyses regarding the relationships between water vapour isotopes and local climate variables, on the other hand, are based on the humidity and corresponding temperature data measured at the meteorological mast.

*Section 3.2.1: Melting and the resulting cutoff of surface temperature at 0 °C is probably less relevant to explain the limited day-to-day variability in 2 m temperature during summer, because melting at Neumayer is mostly a daytime feature, i.e. seldomly lasts a full day, and therefore has less impact on daily mean temperature. A more probable explanation is that because of the absorption of solar radiation the surface radiation deficit becomes small or absent in summer, preventing the formation of strong near-surface temperature inversions in the daily mean. It is the regular formation (clear skies, weak winds) and destruction (cloudy skies, strong winds) of these inversions that explain the large interdiurnal temperature variability in the non-summer seasons.*

We thank the referee for this alternative explanation. We fully agree with these arguments and changed the text accordingly:

Revised manuscript, Subsubsection 3.2.1, line 265:

Daily temperature and $\delta^{18}O$ values in summer are less fluctuating than in the other three seasons (revised manuscript, Fig. 4). This might be explained by a weaker temperature inversion, lower sea ice variability, and stronger sublimation and snowmelt in summer.

Hudson and Brandt (2005) showed that the temperature inversion strength variations in winter are one reason for the large day-to-day variability of 2-meter temperature in Antarctica. In winter, clouds can be much warmer than the surface, which leads to a strong temperature inversion. However, changes in wind speed and direction might change the cloud cover and thereby weaken or destroy the inversion layer, in short time. Due to these processes, stronger temperature inversions can lead to higher temperature variability in winter.

As sea ice can strongly limit the heat flux between a relatively warm ocean and the atmosphere, sea ice coverage variations close to Antarctica's coastal stations can primarily affect the near-surface temperature at the stations (Turner et al., 2020). Decreasing sea ice variability close to the Neumayer Station in summer compared to other seasons, which is true for most other coastal stations in Antarctica, may also lower the temperature variability.

Another reason for the reduced temperature variations in summer can be a stronger heat loss, which prevents temperatures above zero. At Neumayer Station, the largest sources of heat loss in summer are sublimation and snow melting (Jakobs et al., 2019). The sublimation is primarily temperature-controlled and is only significant at Neumayer station in summer. About 19 % of the annual snowfall at this location is removed by sublimation (Van den Broeke et al., 2010). The second source of heat loss at the station is snow melting. In summertime, when the air temperature can rise above 0°C, the surface snow will reach its melting point and start to melt. For the melting process, the incoming radiative energy is partly used for latent heat uptake, keeping the near-surface temperature close to the melting point.

These three phenomena might explain the detected cut-off at 0°C of the 2-m temperature (Fig. 4). They could also partly explain the lower correlation coefficient between the 2-m temperature and $\delta^{18}$O in summer, as upper air temperatures most likely control the latter.

[Figure]

*Figure 3. (Figure 4 in the revised manuscript). $\delta^{18}$O [‰] vs. temperature [°C]. Four plots show daily average temperature–$\delta^{18}$O values for different seasons of the year. For each season, a best fitted line, using the least-squares approach, for $\delta^{18}$O vs. temperature is plotted as a red line and corresponding correlation coefficients are calculated.*

*Fig. 5 and Section 3.2.2: In an environment with unlimited evaporation/sublimation potential such as Neumayer, with near-continuous surface cooling outside of summer, the near-surface air will always be close to saturation (this is what you show in Fig.9). Repeating the correlation of water isotopes with humidity, as you did in Fig. 4 with temperature, is therefore not so useful.*

We agree on this point. Based on another referee's comment we have removed Fig. 5 and Fig. 9 from the revised manuscript, merged the related text, and explain now the link between specific humidity and atmospheric temperature by the dominance of the Clausius–Clapeyron relation:

Revised manuscript, Subsubsection 3.2.1, line 286:

At Neumayer Station, the specific humidity is highly correlated with temperature (Jakobs et al., 2019), as expected from the general Clausius-Clapeyron relation between both quantities. As a consequence, the $\delta^{18}O$ values of water vapour at Neumayer Station are strongly correlated not only to temperature, but also to specific humidity (r = 0.85).

*Section 3.2.3: have you tried correlations with relative humidity?*

We analysed the correlation coefficient for relative humidity, but have not found a high correlation (around -0.35 for relative humidity-$d$ and 0.75 for relative humidity-$\delta^{18}O$).
The correlation coefficient for relative humidity and $\delta^{18}O$ in different seasons are close (spring: r = 0.73, summer: r = 0.57, autumn: r = 0.69, and winter: r = 0.65). There are anti-correlations between the relative humidity and $d$ for spring (r = -0.43), summer (r = -0.59), and autumn (r = -0.19). For winter, there is no correlation between the relative humidity and $d$ (r = 0.04). These findings are added to the revised manuscript:

Revised manuscript, Subsubsection 3.2.2, line 292:

temperature and $\delta^{18}O$. The correlation coefficient for relative humidity and $\delta^{18}O$ in different seasons are similar (spring: $r = 0.73$, summer: $r = 0.57$, autumn: $r = 0.69$, and winter: $r = 0.65$).

Revised manuscript, Subsubsection 3.2.2, line 300:

This pattern can be detected also for temperature-$d$, specific humidity-$d$, and relative humidity-$d$ relations. There is a negative correlation coefficient between temperature and $d$ for spring, $r = -0.41$, summer, r = −0.60, and autumn, r = −0.14, but in winter a weak positive correlation, r = 0.22, is noticed. There are anti-correlations between the specific humidity (relative humidity) values and $d$ for spring, r = −0.50 (r = −0.43), summer, r = −0.71 (r = −0.59), and autumn, r = −0.24 (r = −0.19), which are slightly stronger than the ones between temperature

and *d*. For winter, there is a weak positive correlation between the specific humidity (relative humidity) and *d*, r = 0.13, (r = 0.04).

*Fig. 6: please include the sea ice edge. The difference in latitudinal fetch and absolute uptake appears to be a combination of sea ice extent (sea ice preventing evaporation) and the semi-annual oscillation, the twice-annual expansion/contraction of the circumpolar pressure trough which determines the latitudinal fetch. The absolute temperature also plays an important role, with evaporation/sublimation being reduced at low temperatures (winter).*

We thank the reviewer for this very helpful explanation. We have added the sea ice edge to our analyses (revised manuscript, Subsection 3.3, Figure 5) and this clarifies why in some areas close to the station we do not detect any evaporation:

Revised manuscript, Subsection 3.3, line 310:

Moisture uptake coming to Neumayer Station depends on different factors such as sea ice extent, the Southern Hemisphere semi-annual oscillation (SAO), and absolute temperature. As Fig. 5 shows, the sea ice prevents evaporation from the ocean. In the areas with ice coverage more than 90 %, the moisture uptake is minor. The SAO is the main phenomenon that affects surface pressure changes at the middle and high latitudes of the Southern Hemisphere (Schwerdtfeger, 1967). It means the twice-yearly contraction and compression of the pressure belt surrounding Antarctica as a result of the different heat capacities of the Antarctic continent and the ocean. The SAO leads to a clear half-yearly pressure wave in surface pressure at high latitudes and modifies the atmospheric circulation and temperature cycles (Van Den Broeke, 1998).

Revised manuscript, Subsection 3.3, Figure 5:

[Figure]

*Figure 4. (Figure 5 in the revised manuscript). Simulated mean moisture uptake occurring within the boundary layer [*mm day$^{-1}$*] in the pathway to Neumayer Station during last 10 days modelled by FLEXPART using ECMWF, ERA5 dataset (Hersbach et al., 2020), for spring (SON), summer (DJF), autumn (MMA), and winter (JJA), considering the year of 2017 and 2018. The mean sea ice edge based on ERA5 reanalysis (Hersbach et al., 2020) for ice coverage more than 45% and 90% is shown with light blue and dark blue lines.*

*Fig. 8: This figure shows surface pressure reduced to sea level, and is therefore inaccurate over topography. Values over the continent should be masked.*

The figure is removed because one of other referees asked for it. But, in the revised manuscript (Subsection 2.1, Figure 1), the summer and winter views of this figure are shown as requested in the comments above (Figure 1).

*Section 4.1.1: Temperature at Neumayer is mainly controlled by season (determining the free atmosphere, or 'background' temperature) and surface cooling (determining the negative departure of the near-surface temperature from this background temperature). Eliminating the latter by e.g. selecting the 850 hPa temperature over Neumayer from balloon soundings/ERA5 could facilitate the interpretation of water isotope values in terms of air-masses and large-scale circulation.*

We added the ERA5 850hPa temperature (Hersbach et al., 2020) to our analyses. However, we find that the correlation coefficient between the 850hPa temperature and $\delta^{18}O$ (0.67) is lower than the one between the 2m temperature and $\delta^{18}O$ (0.89). The correlation coefficient is also lower if we look at the different seasons:

| Season | $T_{850hPa}$- $\delta^{18}O$ correlation coefficient | $T_{2m}$- $\delta^{18}O$ correlation coefficient |
|---|---|---|
| Spring | 0.60 | 0.86 |
| Summer | 0.21 | 0.71 |
| Autumn | 0.52 | 0.83 |
| Winter | 0.43 | 0.75 |

For this reason, we decided to continue working with the 2 m temperature in our manuscript, but now also report the lower correlation with the 850hPa temperature:

Revised manuscript, Subsubsection 3.2.1, line 256:

To look at the effect of temperature on isotopic compositions in term of air-masses and large-scale circulation, we examine a higher-level temperature (850 hPa), using ECMWF, ERA5 dataset (Hersbach et al., 2020). The average 850 hPa temperature for the year of 2017 and 2018 is -14.45°C, which is about 1°C warmer than the observed 2-meter average temperature. Daily values of the 850 hPa temperature vary between -31.09°C and -2.97°C, showing a smaller amplitude compared to the 2-meter temperature values at Neumayer Station. For the observational period, the correlation coefficient between the 850 hPa temperature and $\delta^{18}O$ (r = 0.68) is less than the one between the 2-meter temperature and $\delta^{18}O$ (r = 0.89). We can see this characteristic also in a seasonal view. The correlation coefficient for the 850 hPa temperature and $\delta^{18}O$ (the observed 2-meter temperature and $\delta^{18}O$) for different seasons is

calculated: spring: r = 0.60 (r = 0.86); summer: r = 0.21 (r = 0.71); autumn: r = 0.52 (r = 0.83); and winter: r = 0.43 (r = 0.75).

*Minor/textual l.100: "−16.10◦C (±1.05◦C).". What does the uncertainty indicate here? Given measurement uncertainty, suggest removing one digit, i.e. -16.1 °C*

The uncertainty indicates the 1-sigma standard deviation of annual mean temperatures from the long-term mean, calculated for the period 1981-2018 (now explained in the revised manuscript, Subsection 2.1, line 111). The other correction is done as requested:

Revised manuscript, Subsection 2.1, line 108:

The mean annual temperature at Neumayer Station (since 1981) is −16.1 ± 1.1°C (the uncertainty indicates the 1-sigma standard deviation of annual mean temperatures from the long-term mean, calculated for the period 1981-2018).

*l.108: One standard deviation appears rather inclusive. Why was this value selected?*

We used the method defined by Klöwer et al. (2014) (which is referenced in the manuscript). They explain that "The threshold of one standard deviation, which leads to a selection of about 16% of all days as warm (cold) events, is chosen not too high to get a large sample of warm (cold) days."

*l.115: "For the largest time of the year" -> "For most of the year"*

Corrected as suggested.

Revised manuscript, Subsection 2.3, line 125:

For most of the year, wind at Neumayer Station blows from easterly, southerly, or south-westerly directions.

*124: "Cover the whole range..." but the numbers provided fall outside of that range?*

Explained in the revised manuscript:

Subsection 2.3, line 131:

The calibration system at Neumayer Station was modified based on the isotopic composition of water vapour at Neumayer Station and used 3 different isotopic water standards (liquid), with $\delta^{18}$O values of -6.07±0.1 ‰ (around -17 ‰ in water vapour), -25.33±0.1‰ (around -36‰ in water vapour), and -43.80±0.1‰ (around -54 ‰ in water vapour). Water standards with water vapour $\delta^{18}$O values of around -54 ‰, -36 ‰, and -17 ‰, cover the whole isotopic measuremments range in water vapour at Neumayer Station.

and

Revised manuscript, Appendix A, line 597:

The calibration system at Neumayer Station was also modified based on the isotopic composition of water vapour at Neumayer Station and used 3 different isotopic water standards (liquid), with $\delta^{18}O$ values of $-6.07 \pm 0.1$ ‰ (around $-17$ ‰ in water vapour), $-25.33 \pm 0.1$ ‰ (around $-36$ ‰ in water vapour), and $-43.80 \pm 0.1$ ‰ (around $-54$ ‰ in water vapour). $\delta D$ values of the standards (water liquid) are $-43.73 \pm 1.5$ ‰, $-195.21 \pm 1.5$ ‰, and $-344.57 \pm 1.5$ ‰. One of the isotope standards ($\delta^{18}O = -25.33$ ‰) is used for quality control in not one but two of the four bubblers. Every year in January, a sample of each standard is taken and transferred to a laboratory in AWI Bremerhaven and was measured in order to know the isotopic composition. In this study, no change in the isotopic compositions of the standards has been detected. Water standards with water vapour $\delta^{18}O$ values of around $-54$ ‰, $-36$ ‰, and $-17$ ‰, cover the whole isotopic measurements range in water vapour at Neumayer Station.

*l.138: Relative humidity (RH) is commonly expressed as %, not ppm. Moreover, the latter concentration suggests that you are talking about specific humidity (q). Later on you use 'absolute humidity'. Please clarify and provide these numbers (RH and q) for the presented ppm thresholds as well.*

In the revised manuscript, we clarified our wording of different forms of humidity. Now, we distinguish between water concentration given in ppm, specific humidity given in g.kg$^{-1}$, and relative humidity given in %, in the whole text.
We provided specific humidity values for the presented water concentration thresholds in the revised manuscript. But the thresholds cannot be precisely given for relative humidity, since it can be different numbers for the thresholds, depending on the temperature:

Revised manuscript, Subsection 2.3, line 148:

The range of water concentration defined for the Picarro analyser is 1000 to 50000 ppm (parts per million expressed by volume/volume), which equals specific humidity values in the range of $0.62$ to $31.10$ $g\ kg^{-1}$). At Neumayer Station, water concentration easily reaches values below

1000 ppm in the austral winter. For water concentrations lower than 2000 ppm (specific humidity of 1.24 $g\ kg^{-1}$), the analyser shows systematic errors with biases of more than 1 ‰ for $\delta^{18}O$ (Casado et al., 2016).

*l.182: "Humidity": absolute or specific humidity? l.189: "humidity amount..."Check throughout, please.*

As we mentioned above, we clarified all wording related to humidity with respect to water concentration, specific humidity and relative humidity.

*l.222: Only the summer slope differs significantly.*

We agree and removed the sentence related to the highest and lowest slope in the revised manuscript.

*l.262: Check the value of the standard deviation, that appears too large.*

Here we considered the mean value of measurements and the standard deviation is 1-sigma standard deviation of calculating mean value considering hourly averages for all days. We explain it in the revised manuscript, and also consider the daily value instead of the hourly value:

Revised manuscript, Subsection 3.4, line 324:

The origin of the air masses measured at Neumayer Station depends directly on the local wind, which is characterized by relatively high wind speeds, with an annual mean value of 8.7 $m\ s^{-1}$ during the measurement period (with a standard deviation of 5.67 $m\ s^{-1}$, considering daily values of all days).

*l. 302: 10.56 -> 10.6, see also l. 305.*

Corrected as requested:

Revised manuscript, Subsubsection 4.1.2, line 405:

We find that for 64 % of the days (22 out of 36 days) with the wind coming from east and wind speed above the daily average easterly wind of 10.6 $m\ s^{-1}$, the measured $\delta^{18}O$ values are higher than the predicted $\delta^{18}O$ value. This indicates that even for days, when no strong temperature changes can be observed, strong winds from east coincide with more enriched $\delta^{18}O$ values in water vapour at Neumayer Station. On the opposite, for 76 % of the days with katabatic winds and a wind speed higher than the daily averaged southerly wind of 4.6 $m\ s^{-1}$ (12 out of 17 days), measured $\delta^{18}O$ values are lower than the predicted $\delta^{18}O$ values.

*l.309: warm/cold temperature -> high/low temperature*

Corrected as requested (warm/cold event instead of warm/cold temperature):

Revised manuscript, Subsubsection 4.1.2, line 387:

During the observation period, on 86 % of all days that involve warm events at Neumayer Station, the wind came from east.

*l. 324: neither -> either*

Removed from the revised manuscript.

*Section 4.4: suggest changing the title into: Comparison to other sites*

Corrected as requested:

Revised manuscript, Subsection 4.3, line 432:

4.3 Comparison to other sites

*l.366: 'opposite' please rephrase*

Corrected as requested:

Revised manuscript, Subsubsection 4.3.1, line 440:

Most comparable to our measurements is a recent study by Bréant et al. (2019), who have reported water isotopes in vapour from the Dumont d'Urville station in Adélie Land, which is also located at the Antarctic coast (Fig. 1).

*Section 4.4.1: Mention that an extensive ice shelf is absent in the case of DDU, and that the station is situated on an island several km away from the coast.*

Mentioned as requested:

Revised manuscript, Subsubsection 4.3.1, line 442:

During summer, most of the island is free of ice and snow and sea ice is rare (König-Langlo et al., 1998). The average temperature at the Dumont d'Urville station during

We thank referee #3 for his/her detailed comments and suggestions on our manuscript. We hope that we have dealt with all comments in an adequate manner and that the revised manuscript now qualifies for publication in The Cryosphere.

**References**

[revised manuscript text omitted]

---

## Author Response (AR1)

Dear Editor,

Hereby, we would like to submit a revised version of our manuscript and a response to all three referee letters. We greatly appreciate your help in revising this manuscript. The comments of all referees have been comprehensive, practical, and instructive. They have helped us improving the paper.

Regarding your specific comments, we would like to respond as follows:

**Dear Authors,**

First of all, please accept my apologies for the relative length of the procedure for this manuscript. As you know (and as I acted previously) the two initial reviewers agreed on "great data set, poor discussion and interpretation". One of the reviewer suggested to request the opinion of a third reviewer with a specific expertise in Antarctic atmospheric processes. This is what I did, and you have seen the comments of this third reviewer recently posted.

Interactive comment Printer-friendly version Discussion paper As an editor, I was now left with three options: a) request a fourth reviewer's opinion, b) close the discussion and ask for authors final comments and c) reject the paper. Because of the general opinion that you are providing a unique data set, I have decided for option 2.

I believe all reviewers have gone to very detailed and sound comments that would greatly improve the interpretation of the data, and should be fully taken into account in your final response.

Of course, you still have the option to withdraw your manuscript and submit it to another Journal.

We thank you for the opportunity to submit a revised manuscript and continue to believe that The Cryosphere is the appropriate journal for our study. We see the uniqueness of this study in the initial presentation of this new Antarctic water vapour isotope data set. However, we agree that some of our discussions and interpretations have been rather basic, and that additional work could be done for further exploiting the data. Based on the specific comments of the three referees, we think that we have substantially improved the manuscript. We hope that we have dealt with all comments in an adequate manner and that the revised manuscript now qualifies for publication in The Cryosphere.

If you decide that you have enough arguments to answer all reviewers comments, I will ask you to provide us with:

a) A "one to one" reply to all comments of the 3 reviewers, stating your response and how (and where in the manuscript) you have taken this into account in the new version of the manuscript...(citing the initial version of the paragraph and how you have changed it is of real help for the second round of reviews and the editor for his final assessment)

All responses are provided as requested.

b) A pristine version of the new version of the manuscript

As requested, a pristine version of the revised manuscript is available and will be sent to you by email.

c) A new version of the manuscript with changes highlighted

As requested, a version of the manuscript with changes highlighted is also available and will be sent to you by email.

On the basis of these documents, and most probably a second round of review(s), I will take my final decision.

We very much hope that the revised manuscript now meets your expectations and qualifies for publication in The Cryosphere.

**Response to Anonymous Referee #1**

We appreciate the comprehensive, practical, and instructive comments of referee #1. They have helped us improving the paper. We are responding to the comments in the following way:

- The data are presented but the calibration system is not well described and it is questionable to know if it is adapted to the Neumayer conditions. It would be nice to display the calibration curves obtained to correct for the instrumental drift (down to 100 ppm according to the text) as well as the influence of humidity on water isotopic ratio in the humidity range studied here. A section dedicated to calibration should be added in the supplementary material.

We added a subsection called "Calibration program for water vapour isotope measurements" in the supplementary (revised manuscript, Appendix A). Here, we explain all parts of the calibration program and show the humidity response curves:

**Revised manuscript, Appendix A, line 582:**

Appendix A: Calibration program for water vapour isotope measurements

To apply the calibration, we defined a calibration protocol modifying the protocol developed by Steen-Larsen et al. (2013) and Bonne et al. (2014). Based on the protocol we need to correct the isotopic observation in four aspects: the humidity concentration dependence of isotopic measurements, the potential long-term drift of the instrument, the offset between measured and real isotope values, and the wrong measurements related to special events.

To calibrate the instrument, we need to measure different known water stable isotopes in different conditions. One of the modifications of the initial protocol is related to the method of producing water vapour for the calibration. We consider 4 glass bottles (named "bubbler" with a number from 1 to 4) of water standards with known isotopic compositions, kept at a constant temperature. The water vapour with adjustable humidity is made by blowing dry air into the bubblers and making bubbles. For Picarro L2140-i the range of water concentration is originally defined between 1000 and 50000 ppm. For humidity values below 2000 ppm, systematic instrumental errors in the measured isotope values need to be corrected by so-called humidity response functions (see Steen-Larsen et al., 2014b, for details). In order to determine the humidity response functions of our calibration setup, we measure different known isotopic compositions over a range of water concentrations between 100 and 10000 ppm and computed the humidity response functions as the interpolation of the distribution of all measurements for each standard, using a polynomial function of 2nd order (Fig. A1). Humidity response sequences are measured once per year and for this study the humidity response functions of two years (January 2018 and January 2019) have been considered. The calibration system at Neumayer Station was also modified based on the isotopic composition of water vapour at Neumayer Station and used 3 different isotopic water standards (liquid), with  $\delta^{18}$ O values of -6.07±0.1‰ (around -17‰ in water vapour), -25.33±0.1‰ (around -36‰ in water vapour), and

-43.80±0.1 ‰ (around -54 ‰in water vapour). δD values of the standards (water liquid) are --195.21±1.5‰. -344.57±1.5‰. 43.73±1.5‰. and One of the isotope standards  $(\delta^{18}O = -25.33\%)$  is used for quality control in not one but two of the four bubblers. Every year in January, a sample of each standard is taken and transferred to a laboratory in AWI Bremerhaven and was measured in order to know the isotopic composition. In this study, no change in the isotopic compositions of the standards has been detected. Water standards with water vapour  $\delta^{18}$ O values of around -54‰, -36‰, and -17‰, cover the whole isotopic measurements range in water vapour at Neumayer Station. The correction of humidity dependant water vapour isotopic measurements has been done by a linear interpolation of those two moisture response functions that belong to the isotopic standards above and below each measured isotopic value. After applying the humidity response calibration, the correction of the measurements based on the instrumental drift and offset is done. We measure the isotopic composition of all water standards every 25 hours. Such 25-hour cycle avoids repetition of the daily calibration at the same time every day but moves the calibration time 1 hour forward per day. By such a calibration interval, we avoid missing the same time period during all days in all measurements. Each isotopic standard is measured for 30 minutes, and to avoid any memory effects, we consider the mean value of the last 15 minutes of each measured water standard, only. If the calibration measurements are not stable enough, we filter them. To correct the data based on the instrumental drift, we consider the average of 14 days of measurements, 7 days before and 7 days after the day that we want to correct its data. The correction of the instrumental drift and offset are done by a linear interpolation of the two closest mean isotopic standards, measured at the 25-hour interval.

Figure 1. (Figure A1 in the revised manuscript). Humidity response function curves for different standards at different specific humidity levels for  $\delta^{18}O$  and  $\delta D$ . The isotope measurements and their uncertainties (light blue lines) are plotted as anomaly values, calculated in relation to the mean value of the isotopic composition for every bubbler. The values are fitted by a second degree polynomial curve (dark blue line).

- It is not clear why the diurnal cycles are not shown in this study while they are expected to bring interesting information, especially when comparing to other sites as performed in sections 4.4.1 and 4.4.2. It is thus important that the diurnal cycles (at least in summer) are shown and thoroughly discussed. Otherwise, the comparison performed on section 4.4 on the correlation of the isotopic signal does not make sense since they are done on different timescales.

We agree with this suggestion and have now additionally analysed the diurnal cycles, considering two months of two sequent summers (December-January of 2017/18 and 2018/19), in order to compare our data with previously reported studies of the diurnal cycle in Antarctica. We added the subsection "Diurnal cycle" to the paper (revised manuscript, Subsection 3.5). In the Discussion section, we discuss our new results:

Revised manuscript, Subsection 3.5, line 350:

**3.5 Diurnal cycle**

To evaluate the diurnal cycles at Neumayer Station, we consider two months of two sequent summers (December-January of 2017/18 and 2018/19) in order to compare our results with previous studies performed in Antarctica.We derive the daily mean values (the mean of 24 hourly mean values for each day) and subtract it from the time series. The remaining anomalies of all parameters represent an average diurnal cycle (Fig. 8).

The average of all values of each variable for the diurnal cycle study period (December-January of 2017/18 and 2018/19) are:  $\delta^{18}$ O: -26.34 ‰;  $\delta$ D: -205.27 ‰; *d*: 5.46 ‰; 2-meter temperature: -4.25 °C; 10-meter temperature: -3.87 °C; specific humidity: 2.49 g kg-1; relative humidity: 86.30 %; wind speed: 7.53 m s-1; wind direction: 291 degree (we consider only winds with a wind speed of more than 3 m s-1); shortwave downward radiation: 228.98 Wm-2.

Strong diurnal cycles in 2-meter temperature (Fig. 8d, red line), 10-meter temperature (Fig. 8d, green line), specific humidity (Fig. 8e), and relative humidity (Fig. 8f) are detected. For wind speed, the diurnal cycle is weak (Fig. 8g) and for wind direction no diurnal cycle is detectable (Fig. 8h). In summer, there is no strong temperature inversion close to the surface, at least not for the first 10 meters above surface. The temperature differences between 2-meter and 10-meter height reaches up to 1 °C during the coldest time of a day, while during half of the day their difference is less than 0.4 °C. The amplitudes of 10-meter temperature (3.63 °C) are less than 2-meter temperature (4.14 °C). The amplitudes of the specific humidity and relative humidity are 0.54 g kg-1 and 4.19 %, respectively.

A clear diurnal cycle can be detected for  $\delta^{18}$ O (Fig. 8a),  $\delta$ D (Fig. 8b), and *d* (Fig. 8c). The diurnal amplitudes are 2.45 ‰, 21.07 ‰, and 4.87 ‰, respectively. A very high correlation coefficient between  $\delta^{18}$ O and 2-meter temperature (r = 0.98) and 10-meter temperature (r = 0.99) suggests the temperature changes as the main driver of water vapour  $\delta^{18}$ O diurnal variations. *d* is rather anti-correlated with relative humidity (r = -0.59), while it does not show a considerable correlation with temperature and specific humidity.

The 2 and 10-meter temperature cycles and consequently  $\delta^{18}O$  and  $\delta D$  follow the shortwave radiation (Fig. 8i) with a short delay (around 3 hours) and show the minimum and maximum values at 03:00 UTC (local time) and 15:00 UTC. The relative humidity behaves the other way

round and shows the minimum value at 15:00 UTC and and maximum values between 21:00 UTC and 03:00 UTC. *d* has the minimum at 00:00 UTC and the maximum at 09:00 UTC.

---

## Author Response (AR2)

Dear Editor,

Hereby, we would like to submit a revised version of our manuscript and a response to the referee letter. We hope that the revised manuscript now is qualified for publication in The Cryosphere.

**Response to Anonymous Referee #3**

We appreciate the comments of referee #3. We are responding to the comments in the following way:

*l. 87: "The station is situated on the 200-meter thick Ekström ice shelf" This is the local ice thickness, ice shelf thickness typically increases strongly towards the grounding line.*

Corrected as requested. Revised manuscript, Subsubsection 2.1, line 87:

The station is locally situated on a 200-meter-thick ice shelf (Ekström) approx. 42 m above sea level. This ice shelf with the thickness that increases strongly towards the grounding line, has a homogeneous, flat surface slightly sloping upwards to the south (Klöwer et al., 2013).

*l. 98: "These low-pressure systems move toward west around Antarctica" Generally, low-pressure systems move eastwards with the westerly circulation around Antarctica.*

You are right, thanks. We correct the sentence. Revised manuscript, Subsubsection 2.1, line 99:

These low-pressure systems move towards east around Antarctica with cyclonic circulations (clockwise).

*l. 113: "In this study, temperature, relative and specific humidity…" Temperature and relative humidity are measured directly, after which specific humidity is then calculated, also using pressure.*

Corrected as requested. Revised manuscript, Subsubsection 2.2, line 114:

In this study, temperature, relative humidity, pressure, wind speed, and wind direction data, measured 50 meters away from the station at 2-meter height above the surface, are used. Additionally, we calculated specific humidity based on the measured relative humidity, temperature, and pressure.

*l. 228: "Daily relative humidity fluctuates between 59.95 % and 96.71 % with a mean value of 80.42 %." Suggest changing to "Daily relative humidity fluctuates between 60 % and 97 % with a mean value of 80 %."*

Corrected as requested. Revised manuscript, Subsubsection 3.1, line 230:

Daily relative humidity fluctuates between 60 % and 97 % with a mean value of 80 %.

*l. 232: "-15.81∘C and -15.65" -> "-15.8∘C and -15.7"*

Corrected as requested. Revised manuscript, Subsubsection 3.1, line 233:

The average 2-m temperature for the year 2017 and 2018 is -15.8℃ and -15.7℃, respectively.

*l. 258-260: Use one decimal for temperature.*

Corrected as requested. Revised manuscript, Subsubsection 3.2.1, line 259:

The average 850 hPa temperature for the year of 2017 and 2018 is -14.5 ℃, which is about 1 ℃ warmer than the observed 2-meter average temperature. Daily values of the 850 hPa temperature vary between -31.1 ℃ and -3.0 ℃, showing a smaller amplitude compared to the 2-meter temperature values at Neumayer Station.

*l. 268: "In winter, clouds can be much warmer than the surface, which leads to a strong temperature inversion. However, changes in wind speed and direction might change the cloud cover and thereby weaken or destroy the inversion layer, in short time."*

*Suggest to change to:*

[revised manuscript text omitted]

*l. 355 -> 365: for the meteorological variables, use a single decimal. Do this throughout.*

Corrected as requested. Revised manuscript, Subsubsection 3.5, line 355:

The average of all values of each variable for the diurnal cycle study period (December-January of 2017/18 and 2018/19) are: $\delta^{18}$O: −26.3 ‰; $\delta$D: −205.3 ‰; d: 5.5 ‰; 2-meter temperature: −4.3 ℃; 10-meter temperature: −3.9 ℃; specific humidity: 2.5 g kg$^{-1}$; relative humidity: 86.3 %; wind speed: 7.6 m s$^{-1}$; wind direction: 291 degree (we consider only winds with a wind speed of more than 3 m s$^{-1}$); shortwave downward radiation: 229.0 W m$^{-2}$.
Strong diurnal cycles in 2-meter temperature (Fig. 8d, red line), 10-meter temperature (Fig. 8d, green line), specific humidity (Fig. 8e), and relative humidity (Fig. 8f) are detected. For wind speed, the diurnal cycle is weak (Fig. 8g) and for wind direction no diurnal cycle is detectable (Fig. 8h). In summer, there is no strong temperature inversion close to the surface, at least not for the first 10 meters above surface. The temperature differences between 2-meter and 10-meter height reaches up to 1 ℃ during the coldest time of a day, while during half of the day their difference is less than 0.4 ℃.

*l. 367: "A very high correlation coefficient between $\delta^{18}$O and 2-meter temperature (r = 0.98) and 10-meter temperature (r = 0.99) suggests the temperature changes as the main driver of water vapour $\delta^{18}$O diurnal variations" The fact that both variables show a daily cycle will*

*result in high correlation but does not prove causation. Could be solar radiation, sublimation, boundary layer depth….this also applies to similar statements elsewhere in the paper.*

According to the ultimate goal of studying isotopes, we have changed the argument. Revised manuscript, Subsubsection 3.5, line 367:

There is a very high correlation coefficient between $\delta^{18}O$ and 2-meter temperature (r = 0.98) and 10-meter temperature (r = 0.99). These high correlations cause the temperature to be predictable based on water stable isotopes variations. *d* is rather anti-correlated with relative humidity (r = −0.59), while it does not show a considerable correlation with temperature and specific humidity.

*l. 398: I could be wrong, but it appears that Fig. 10 is referenced before Fig. 9.*

You are right; thank you for the remark. We have reordered Fig. 9 and Fig. 10.

*l. 402: pattern -> patterns*

Corrected as requested. Revised manuscript, Subsubsection 4.1.2, line 401:

Then we look at the residual between the predicted $\delta^{18}O$ value and the observed $\delta^{18}O$ value (Fig. 10) and analyse how this residual might be linked to different wind patterns at Neumayer Station.

We thank referee #3 for his/her detailed comments and suggestions on our manuscript. We hope that we have dealt with all comments in an adequate manner and that the revised manuscript now qualifies for publication in The Cryosphere.

---

## Author Response (AR3)

Dear Editor,

Hereby, we would like to submit a revised version of our manuscript and a response to your comments in the following way:

*Thank you very much for this corrected manuscript along the lines of the referee. You indeed answered to most of it. I am still left with two items:*
*a) The reviewer specifically asked that all meteorological variables are mentioned with only one decimal.. he says "do this throughout".. but I still found some with two decimals!.. can you perform this full screening of your manuscript to be sure none is left over... thank you*

Corrected as requested. Now all meteorological variables are mentioned with only one decimal.

*b) The referee still regrets that you use strong correlation as "causality".. you changed that in the place he mentions it (d18O vs. Temperatures), but the reviewer suggests that you might have made similar statement elsewhere in the paper.. please ensure that this is not the case and correct accordingly if it is the case!..*

We agree that we should not mix correlation and causality. Therefore, we checked the text and deleted a part that created this ambiguity.

"Relatively high correlations between daily $\delta^{18}O$ and temperature (r=0.89), and also between daily $\delta^{18}O$ and specific humidity (r = 0.85), highlight the role of temperature and humidity as the main drivers of $\delta^{18}O$ variations in water vapour." Is removed (Revised manuscript, Subsubsection 4.1.1, line 378).

We hope that the revised manuscript now is qualified for publication in The Cryosphere.